# Oligodendrocyte-lineage cell exocytosis and L-type prostaglandin D synthase promote oligodendrocyte development and myelination

Lin Pan[1], Amelia Trimarco[2], Alice J Zhang[1], Ko Fujimori[3], Yoshihiro Urade[4], Lu O Sun[5], Carla Taveggia[2]*, Ye Zhang[1,6,7,8]*

[1]Department of Psychiatry and Biobehavioral Sciences, Intellectual and Developmental Disabilities Research Center, Semel Institute for Neuroscience and Human Behavior, David Geffen School of Medicine, University of California, Los Angeles, Los Angeles, United States; [2]Division of Neuroscience, IRCCS, San Raffaele Hospital, Milan, Italy; [3]Department of Pathobiochemistry, Osaka Medical and Pharmaceutical University, Osaka, Japan; [4]Hirono Satellite, Isotope Science Center, The University of Tokyo, Fukushima, Japan; [5]Department of Molecular Biology, University of Texas Southwestern Medical Center, Dallas, United States; [6]Brain Research Institute, University of California, Los Angeles, Los Angeles, United States; [7]Eli and Edythe Broad Center of Regenerative Medicine and Stem Cell Research, University of California, Los Angeles, Los Angeles, United States; [8]Molecular Biology Institute, University of California, Los Angeles, Los Angeles, United States

*For correspondence:
taveggia.carla@hsr.it (CT);
yezhang@ucla.edu (YZ)

**Abstract** In the developing central nervous system, oligodendrocyte precursor cells (OPCs) differentiate into oligodendrocytes, which form myelin around axons. Oligodendrocytes and myelin are essential for the function of the central nervous system, as evidenced by the severe neurological symptoms that arise in demyelinating diseases such as multiple sclerosis and leukodystrophy. Although many cell-intrinsic mechanisms that regulate oligodendrocyte development and myelination have been reported, it remains unclear whether interactions among oligodendrocyte-lineage cells (OPCs and oligodendrocytes) affect oligodendrocyte development and myelination. Here, we show that blocking vesicle-associated membrane protein (VAMP) 1/2/3-dependent exocytosis from oligodendrocyte-lineage cells impairs oligodendrocyte development, myelination, and motor behavior in mice. Adding oligodendrocyte-lineage cell-secreted molecules to secretion-deficient OPC cultures partially restores the morphological maturation of oligodendrocytes. Moreover, we identified L-type prostaglandin D synthase as an oligodendrocyte-lineage cell-secreted protein that promotes oligodendrocyte development and myelination in vivo. These findings reveal a novel autocrine/paracrine loop model for the regulation of oligodendrocyte and myelin development.

## Editor's evaluation

The manuscript will be of interest to glial and myelin disease researchers. The well-designed combination of in vitro and in vivo approaches uncovers a potential mechanism of autocrine/paracrine signaling in oligodendrocyte maturation which provides an exciting avenue for future investigation.

## Introduction

In the developing central nervous system (CNS), oligodendrocyte precursor cells (OPCs) differentiate into oligodendrocytes (*Bergles and Richardson, 2016*; *Hill et al., 2014*; *Kang et al., 2010*), which form myelin sheaths around axons. Myelin is essential for the propagation of action potentials and for the metabolism and health of axons (*Fünfschilling et al., 2012*; *Larson et al., 2018*; *Mukherjee et al., 2020*; *Saab et al., 2016*; *Schirmer et al., 2018*; *Simons and Nave, 2016*). When oligodendrocytes and myelin are damaged in demyelinating diseases such as multiple sclerosis (MS) and leukodystrophy, sensory, motor, and cognitive deficits can ensue (*Gruchot et al., 2019*; *Lubetzki et al., 2020*; *Stadelmann et al., 2019*). In a broader range of neurological disorders involving neuronal loss, such as brain/spinal cord injury and stroke, the growth and myelination of new axons are necessary for neural repair (*Wang et al., 2020*). Thus, understanding oligodendrocyte development and myelination is critical for developing treatments for a broad range of neurological disorders.

Over the past several decades, researchers have made great progress in elucidating the cell-intrinsic regulation of oligodendrocyte development and myelination (e.g. transcription factors, epigenetic mechanisms, and cell death pathways) (*Aggarwal et al., 2013*; *Bergles and Richardson, 2016*; *Budde et al., 2010*; *Dugas et al., 2010*; *Elbaz and Popko, 2019*; *Elbaz et al., 2018*; *Emery and Lu, 2015*; *Emery et al., 2009*; *Fedder-Semmes and Appel, 2021*; *Foerster et al., 2020*; *Harrington et al., 2010*; *Herbert and Monk, 2017*; *Howng et al., 2010*; *Koenning et al., 2012*; *Mitew et al., 2018*; *Nawaz et al., 2015*; *Snaidero et al., 2017*; *Sun et al., 2018*; *Wang et al., 2017*; *Xu et al., 2020*; *Zhao et al., 2018*; *Zuchero et al., 2015*), as well as the cell-extrinsic regulation by other cell types (e.g. neurons (*Gibson et al., 2014*; *Hines et al., 2015*; *Mayoral et al., 2018*; *Osso et al., 2021*; *Redmond et al., 2016*; *Wake et al., 2011*), microglia/macrophages (*Butovsky et al., 2006*; *Sherafat et al., 2021*), and lymphocytes *Dombrowski et al., 2017*). However, it remains unclear whether interactions among oligodendrocyte-lineage cells (OPCs and oligodendrocytes) affect oligodendrocyte development and myelination.

One of the most abundant transcripts encoding secreted proteins in oligodendrocyte-lineage cells is *Ptgds*, encoding lipocalin-type prostaglandin D synthase (L-PGDS; *Zhang et al., 2014*; *Zhang et al., 2016*). Oligodendrocytes and meningeal cells are major sources of L-PGDS in the CNS (*Urade et al., 1993*; *Urade, 2021*; *Zhang et al., 2014*; *Zhang et al., 2016*). L-PGDS has two functions: as an enzyme and as a carrier (*Urade and Hayaishi, 2000*, *Urade, 2021*). As an enzyme, L-PGDS converts prostaglandin H2 to prostaglandin D2 (PGD2) (*Urade et al., 1985*). PGD2 regulates sleep, pain, and allergic reactions (*Eguchi et al., 1999*; *Satoh et al., 2006*; *Urade and Hayaishi, 2011*). L-PGDS also binds and transports lipophilic molecules such as thyroid hormone, retinoic acid, and amyloid-β (*Urade and Hayaishi, 2000*) and promotes Schwann cell myelination in the peripheral nervous system (*Trimarco et al., 2014*). Yet, its function in the development of the CNS is unknown.

To determine whether cell-cell interactions within the oligodendrocyte lineage regulate oligodendrocyte development, we blocked VAMP1/2/3-dependent exocytosis from oligodendrocyte-lineage cells in vivo and found impairment in oligodendrocyte development, myelination, and motor behavior in mice. Similarly, exocytosis-deficient OPCs exhibited impaired development in vitro. Adding oligodendrocyte-lineage cell-secreted molecules promoted oligodendrocyte development. These results suggest that an autocrine/paracrine loop promotes oligodendrocyte development and myelination. We assessed L-PGDS as a candidate autocrine/paracrine signal and further discovered that oligodendrocyte development and myelination were impaired in L-PGDS-knockout mice. Moreover, overexpression of the gene encoding L-PGDS partially restored the myelination defect of exocytosis-deficient mice. These results reveal a new autocrine/paracrine loop model for the regulation of oligodendrocyte development in which VAMP1/2/3-dependent exocytosis from oligodendrocyte-linage cells and secreted L-PGDS promote oligodendrocyte development and myelination.

## Results

### Expression of botulinum toxin B in oligodendrocyte-lineage cells in vivo

If oligodendrocyte-lineage cells use autocrine/paracrine mechanisms to promote development and myelination, one would predict that (1) blocking secretion from oligodendrocyte-lineage cells would impair oligodendrocyte development and myelination and, in turn, that (2) adding oligodendrocyte-lineage cell-secreted molecules might promote oligodendrocyte development. Membrane fusion

relies on soluble N-ethylmaleimide-sensitive fusion protein attachment protein receptors (SNARE) family proteins located on vesicles (v-SNAREs) and target membranes (t-SNAREs). The binding of v-SNAREs and t-SNARES form intertwined α-helical bundles that generate force for membrane fusion (*Pobbati et al., 2006*). VAMP1/2/3 are v-SNAREs that drive the fusion of vesicles with the plasma membrane to mediate exocytosis (*Chen and Scheller, 2001*). We found that oligodendrocyte-lineage cells express high levels of VAMP2 and VAMP3 and low levels of VAMP1 in vivo (*Figure 1A–C*; *Zhang et al., 2014*; *Zhang et al., 2016*), consistent with previous reports in vitro (*Feldmann et al., 2009*; *Feldmann et al., 2011*; *Madison et al., 1999*). We found that VAMP2$^+$ and VAMP3$^+$ puncta are distributed throughout cultured OPCs, including.

In the soma and processes (*Figure 1—figure supplement 1*). Botulinum toxin B specifically cleaves VAMP1/2/3 (*Yamamoto et al., 2012*), but not VAMP4, 5, 7, or 8 (*Yamamoto et al., 2012*), and inhibits the release of vesicles containing proteins (*Somm et al., 2012*) as well as small molecules such as neural transmitters (*Poulain et al., 1988*). Of note, botulinum toxin B does not cleave VAMP proteins that are involved in the vesicular transport between the trans-Golgi network, endosomes, and lysosomes (*Antonin et al., 2000*; *Hoai et al., 2007*; *Pols et al., 2013*). Similarly, botulinum toxins do not affect ion channel- or membrane transporter-mediated release of small molecules.

To block VAMP1/2/3-dependent exocytosis from oligodendrocyte-lineage cells, we crossed *Pdgfra-CreER* transgenic mice, which express Cre recombinase in OPCs (PDGFRα$^+$Olig2$^+$; *Kang et al., 2010*), with loxP-stop-loxP-botulinum toxin B light chain-IRES-green fluorescent protein (GFP) (inducible botulinum toxin B, or ibot) transgenic mice (*Slezak et al., 2012*), allowing expression of botulinum toxin B-light chain in OPCs and their progeny. The light chain contains the catalytically active domain of the toxin but lacks the heavy chain, which allows cell entry (*Montal, 2010*), thus confining toxin expression to the targeted cell type. Therefore, the ibot transgenic mice allow for the inhibition of VAMP1/2/3-dependent exocytosis in a cell-type-specific and temporally controlled manner (*Slezak et al., 2012*).

In our study, we used double-transgenic mice hemizygous for both Cre and ibot and referred to them as the PD:ibot mice thereafter. To validate our model and test its recombination efficiency, we injected 0.1 mg of 4-hydroxytamoxifen in each PD:ibot mouse daily for 2 days between postnatal days 2–4 (P2-4) and examined GFP expression at P8 and P30. We assessed whether GFP expression is restricted to oligodendrocyte-lineage cells (specificity) and what proportion of oligodendrocyte-lineage cells express GFP (efficiency/coverage). At P8, when the vast majority of oligodendrocyte-lineage cells are undifferentiated OPCs, we detected specific expression of GFP in oligodendrocyte-lineage cells (*Figure 1D–F*). GFP was efficiently expressed by oligodendrocyte-lineage cells throughout the brain, including the cerebral cortex grey matter, corpus callosum, and striatum (*Figure 1G*). At P30, when substantial numbers of OPCs have differentiated (PDGFRα$^-$Olig2$^+$), we observed a similarly high specificity of GFP expression in oligodendrocyte-lineage cells (*Figure 1—figure supplement 2*). These observations are consistent with previous reports on the specificity and efficiency of the *Pdgfra-CreER* transgenic line (*Kang et al., 2010*). As controls, we used wildtype mice, mice with only the Cre transgene or only the ibot transgene subjected to the same tamoxifen injection scheme. In all control conditions, we detected very little GFP expression (*Figure 1—figure supplement 3*).

To directly assess the expression of botulinum toxin B-light chain and the cleavage of the VAMP proteins in oligodendrocyte-lineage cells from PD:ibot mice, we purified OPCs from PD:ibot and control mice by immunopanning and allowed them to differentiate into oligodendrocytes in culture. We performed western blot analysis of the cultures and detected botulinum toxin B-light chain in PD:ibot but not in control cells (*Figure 1H*). Furthermore, the levels of full-length VAMP2 and VAMP3 proteins were lower in PD:ibot cells compared with control cells (*Figure 1I, J, L and M*). Based on these observations, we conclude that the botulinum toxin-GFP transgene is specifically and efficiently expressed by oligodendrocyte-lineage cells in PD:ibot mice.

## Blocking VAMP1/2/3-dependent exocytosis from oligodendrocyte-lineage cells impairs oligodendrocyte development, myelination, and motor behavior

In PD:ibot mice, we found that the numbers of differentiated oligodendrocytes (CC1$^+$) were reduced in the cerebral cortex at P11, 15, and 30 (*Figure 2A and B*), whereas the numbers of OPCs (PDGFRα$^+$) did not change (*Figure 2C, E, F and H*). Olig2 labels both OPCs and differentiated oligodendrocytes,

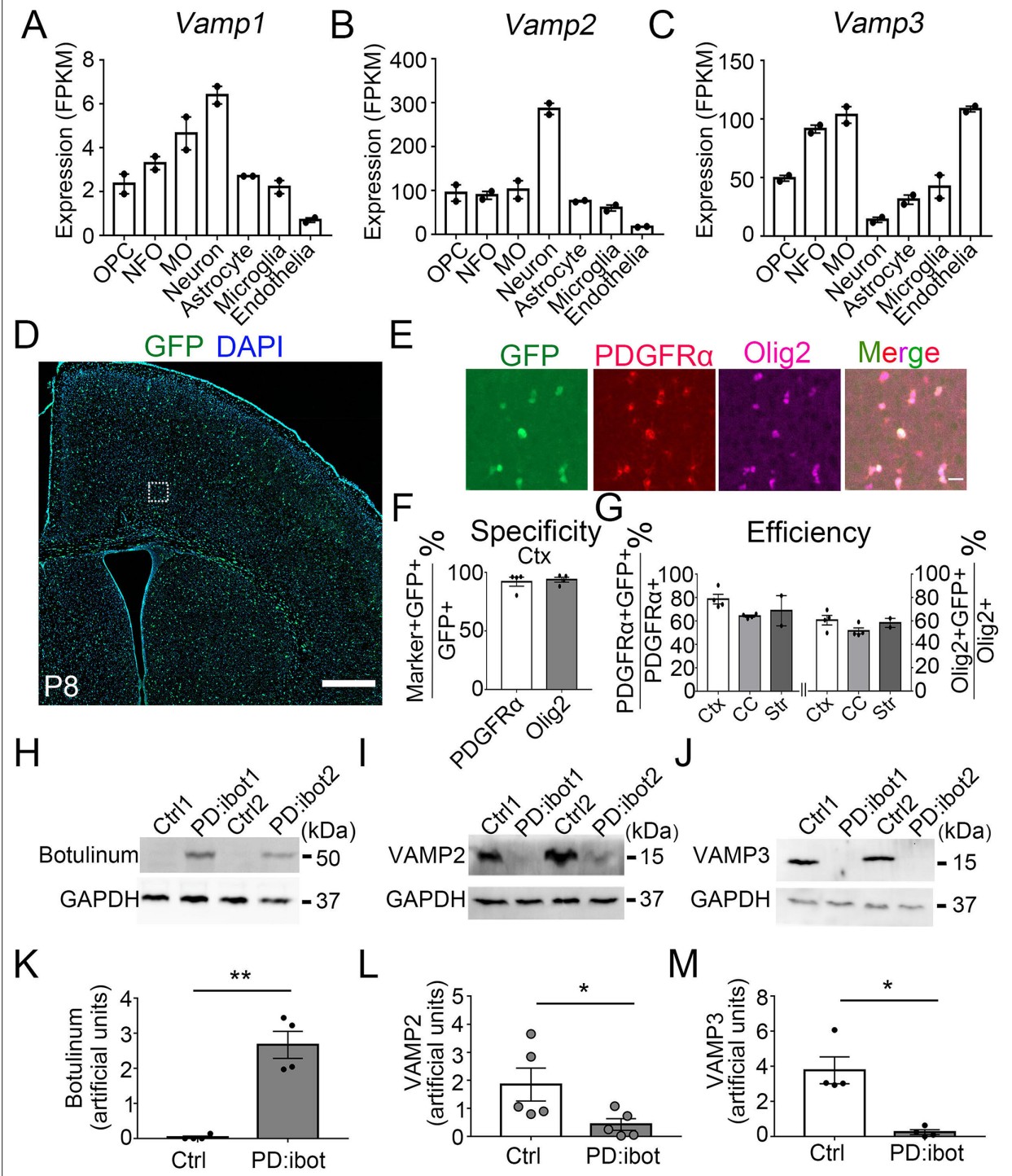

**Figure 1.** VAMP1/2/3 and ibot expression in oligodendrocyte-lineage cells. (**A–C**) Expression of VAMP1/2/3 by oligodendrocyte-lineage cells, neuron, astrocyte, microglia, and endothelia determined by RNA-seq (*Zhang et al., 2014*). NFO, newly formed oligodendrocytes. MO, myelinating oligodendrocytes. (**D**) Expression of ibot-GFP in *Pdgfra-CreER*; ibot (PD:ibot) mice. Scale bar: 500 µm. (**E**) Colocalization of ibot-GFP with PDGFRα and Olig2 in PD:ibot mice. Scale bar: 20 µm. This image is from box area in (**D**). (**F**) Specificity of ibot-GFP expression in oligodendrocyte-lineage cells in cortex. N=4 mice per group. 92.1 ± 3.9% of GFP⁺ cells were PDGFRα⁺; 93.8 ± 2.1% of GFP⁺ cells were Olig2⁺. (**G**) Efficiency of ibot-GFP expression in oligodendrocyte-lineage cells in cortex, corpus callosum, and striatum. N=4 mice per group for cortex and corpus callosum. N=2 mice per group for striatum. In cortex, 78.7 ± 4.1% of PDGFRα⁺ cells were GFP⁺; 60.6 ± 4.1% of Olig2⁺ cells were GFP⁺. In corpus callosum, 64.1 ± 0.9% of PDGFRα⁺ cells were GFP⁺; 51.5 ± 2.6% of Olig2⁺ cells were GFP⁺. In striatum, 68.8 ± 12.9% of PDGFRα⁺ cells were GFP⁺; 58.4 ± 3.9% of Olig2⁺ cells were GFP⁺. P8 mice were used in (**D–G**). (**H**) Presence of botulinum toxin B-light chain in oligodendrocyte cultures from 4-hydroxytamoxifen-injected PD:ibot mice detected

*Figure 1 continued on next page*

*Figure 1 continued*

by western blots. (**I**) Reduced levels of full-length VAMP2 in oligodendrocyte cultures from 4-hydroxytamoxifen-injected PD:ibot mice determined by Western blots. (**J**) Reduced levels of full-length VAMP3 in oligodendrocyte cultures from 4-hydroxytamoxifen-injected PD:ibot mice determined by western blots. (**K**) Quantification of botulinum toxin B-light chain immunoblot signal intensity. N=4 mice per group. Paired two-tailed T-test. *, p<0.05. **, p<0.01. ***, p<0.001. NS, not significant. Botulinum toxin B-light chain intensity: 0.03±0.03 in ctrl and 2.7±0.4 in PD:ibot, p=0.008. (**L**) Quantification of VAMP2 immunoblot signal intensity. N=5 mice per group. Paired two-tailed T-test. VAMP2 intensity: 1.8±0.6 in ctrl and 0.4±0.2 in PD:ibot, p=0.03. (**M**) Quantification of VAMP3 immunoblot signal intensity. N=4 mice per group. VAMP3 intensity: 3.8±0.8 in ctrl and 0.2±0.1 in PD:ibot, p=0.01.

The online version of this article includes the following source data and figure supplement(s) for figure 1:

**Source data 1.** It contains original data points for *Figure 1A, B, C, F, G, K, L and M*.

**Source data 2.** It contains figures with the uncropped blots with the relevant bands clearly labelled (**A**) and original blot images (**B–G**).

**Figure supplement 1.** VAMP2+ and VAMP3+ puncta are distributed throughout cultured OPCs, including soma and processes.

**Figure supplement 2.** Specificity and efficiency of ibot expression in oligodendrocyte-lineage cells at P30.

**Figure supplement 2—source data 1.** It contains original data points for *Figure 1—figure supplement 2B and C*.

**Figure supplement 3.** Distribution of GFP+ cells in the brain at P8.

**Figure supplement 3—source data 1.** It contains original data points for *Figure 1—figure supplement 3B and C*.

and the densities of Olig2+ cells in PD:ibot mice were also reduced, likely due to the reduction in differentiated oligodendrocytes (*Figure 2C, D, F and G*). At P8, the vast majority of PDGFRα-CreER-expressing cells are OPCs (*Paukert et al., 2014*). Therefore, it is more likely that blocking exocytosis from OPCs rather than oligodendrocytes affects oligodendrocyte development during the early post-natal period.

To determine the stage(s) of oligodendrocyte development affected by botulinum toxin, we performed RNAscope in situ hybridization in vivo using probes for *Enpp6*, a marker for pre-myelinating oligodendrocytes, and *Mbp*, a marker for oligodendrocytes. Interestingly, both markers showed a significant reduction in PD:ibot mice compared with controls, demonstrating that the oligodendrocyte development defect in PD:ibot mice manifest as early as the pre-myelinating stage (*Figure 2I, J and K*).

We next examined myelin development in PD:ibot mice and found that immunofluorescence of myelin basic protein (MBP), one of the main components of CNS myelin, is reduced in PD:ibot mice (*Figure 3A–H*). Moreover, many MBP+ ibot-GFP–expressing cells exhibit round cell morphology whereas MBP+GFP– control cells form elongated myelin internodes along axon tracks (*Figure 3I*). Transmission electron microscopy allows for the assessment of myelin structure at a high resolution. Thus, to further examine myelination in PD:ibot mice, we performed transmission electron microscopy imaging and found a reduction in the percentage of myelinated axons (Fig. 3 J, L) and reduced myelin thickness in PD:ibot mice (*g-ratio*: axon diameter divided by the diameter of axon +myelin; *Figure 3K and M*). The density and diameter of axons did not differ between PD:ibot and control mice (*Figure 3N and O*).

To determine whether the reduction of oligodendrocytes in PD:ibot mice is caused by cell death, we performed immunostaining with an antibody against cleaved caspase-3, which labels apoptotic cells. We observed no difference in the total apoptotic cells (cleaved caspase-3+), apoptotic OPCs (cleaved caspase-3+PDGFRα+), apoptotic oligodendrocyte-lineage cells (cleaved caspase-3+Olig2+), or apoptotic cells from other lineages (cleaved caspase-3+Olig2–) between PD:ibot and control mice in the cerebral cortex at P8, 15 and 30 (*Figure 3—figure supplement 1*).

To determine whether botulinum toxin-B-expressing cells contribute to the population of surviving differentiated oligodendrocytes, we examined the overlap between GFP+ botulinum-expressing cells and differentiated oligodendrocytes (Olig2+PDGFRα– cells and CC1+ cells) and found that botulinum-expressing cells can survive and become differentiated oligodendrocytes at P8 and P30 (*Figure 3—figure supplement 2A*). We next compared the ratio of CC1+ cells to Olig2+ cells in GFP+ cells in PD:ibot mice, GFP– cells in PD:ibot mice, and all cells (GFP–) in control cells and did not observe statistically significant difference between any of the groups (*Figure 3—figure supplement 2B*). To examine whether OPC proliferation is affected in PD:ibot mice, we quantified the percentage of Ki-67+ proliferating cells among PDGFRα+ OPCs and found an increase in OPC proliferation (*Figure 3—figure supplement 3*).

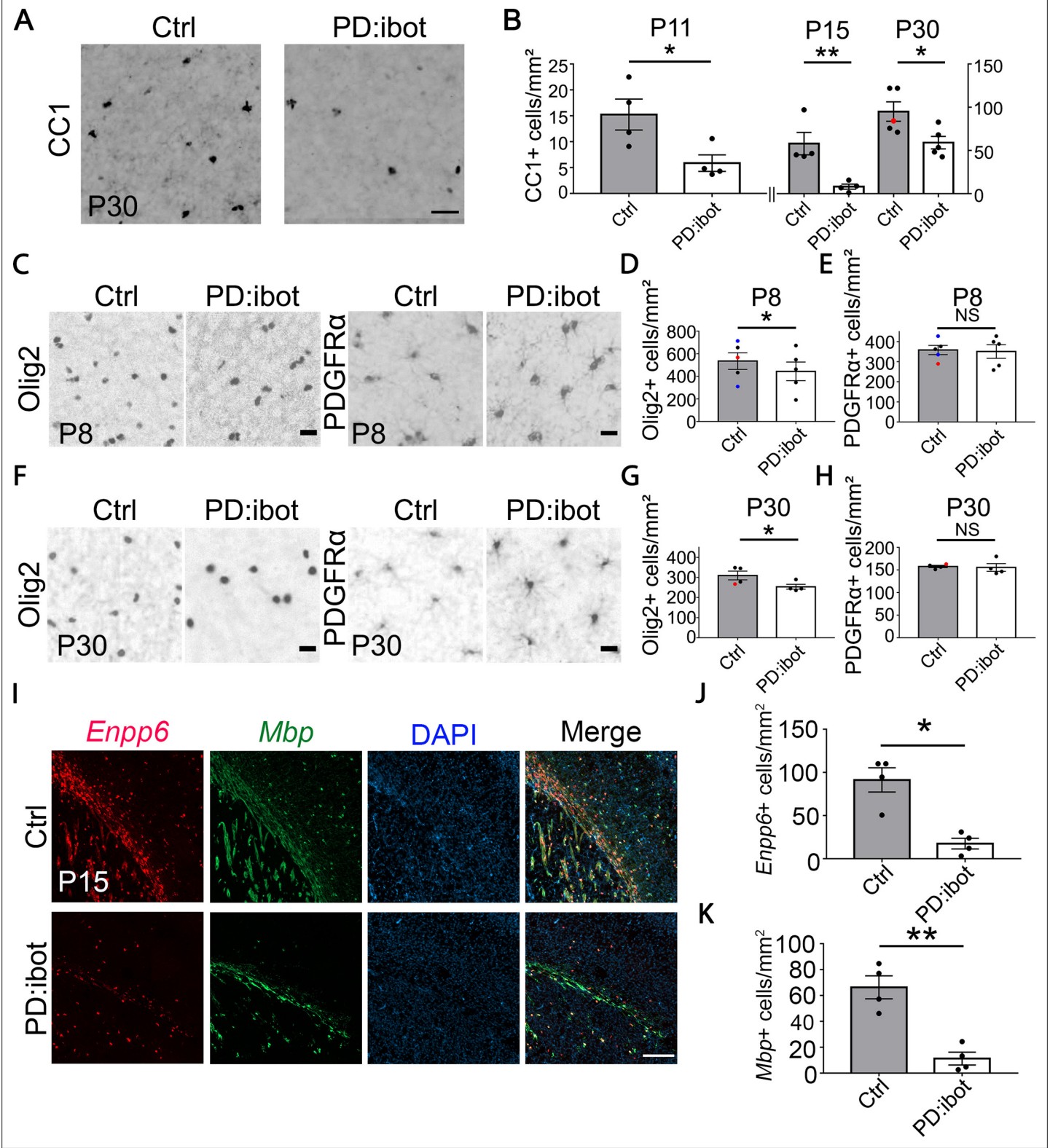

**Figure 2.** Reduction of CC1[+], Olig2[+], *Enpp6*[+], and *Mbp*[+] oligodendrocytes in PD:ibot mice. (**A**) Differentiated oligodendrocytes labeled by CC1 in the cerebral cortex of PD:ibot and control mice at P30. Scale bar: 50 µm. (**B**) Quantification of the density of CC1[+] differentiated oligodendrocytes in the cerebral cortex of PD:ibot and control mice at P11, 15, and 30. N=4 mice per group at P11 and P15. N=5 mice per group for P30. The genotype of control mice is indicated by color: black dots for ibot only and red dots for Cre only. Paired two-tailed T-test. CC1[+] cells/mm²: 15.3±3.0 in control and 5.9±1.6 in PD:ibot, p=0.046 at P11. 57.9±13 in control and 8.1±3.0 in PD:ibot, p=0.018 at P15. 95.0±11.2 in control and 59.0±7.3 in PD:ibot, p=0.049

*Figure 2 continued on next page*

*Figure 2 continued*

at P30. (**C, F**) Olig2⁺ oligodendrocyte-lineage cells and PDGFRα⁺ OPCs in the cerebral cortex of PDibot and control mice at P8 (**C**) and P30 (**F**). Scale bars: 20 µm. (**D, E, G, H**) Quantification of the densities of Olig2⁺ and PDGFRα⁺ cells in the cerebral cortex of PD:ibot and control mice at P8 and P30. N=5 mice per group at P8. N=4 mice per group at P30. Paired two-tailed T-test. Olig2⁺ cells/mm²: 535.6±73.6 in control and 444.4±82.9 in PD:ibot, p=0.040 at P8; 309.9±21.7 in control and 253.8±12.2 in PD:ibot, p=0.048 at P30. PDGFRα⁺ cells/mm²: 358.4±22.8 in control and 350.9±34.0 in PD:ibot, p=0.74 at P8; 157.8±2.6 in control and 155.6±8.5 in PD:ibot, p=0.85 at P30. (**I**) *Enpp6*⁺ cells and *Mbp*⁺ cells in the brains of PDibot and control mice at p15. Scale bar: 200 µm. (**J, K**) Quantification of the density of *Enpp6*⁺ cells and *Mbp*⁺ cells in the cerebral cortex of PD:ibot and control mice at P15. N=4 mice per group. All control mice are ibot only (black dots). Paired two-tailed T-test. *Enpp6*⁺ cells/mm²: 91.4±14.1 in control and 17.5±6.2 in PD:ibot, p=0.015; *Mbp*⁺ cells/mm²: 66.3±8.9 in control and 11.3±4.9 in PD:ibot, p=0.0057.

The online version of this article includes the following source data for figure 2:

**Source data 1.** It contains original data points for *Figure 2B, D, E, G, H, J and K*.

To investigate whether the expression of botulinum toxin B-light chain affects oligodendrocyte development and myelination in non-cell-type-specific manners, we blocked exocytosis from astrocytes or endothelial cells by crossing ibot transgenic mouse with *Gfap*-Cre (line 77.6) and *Tek*-Cre strains, respectively. We found astrocyte- and endothelial cell-specific expression of ibot-GFP in these mice but did not detect any obvious changes in oligodendrocyte density or myelin proteins. These observations suggest that botulinum toxin B-light chain peptides have specific effects on the targeted cell types. However, we could not rule out the possibility that embryonic and/or early postnatal compensation may mask the effect when a constitutive Cre is used.

To examine the functional consequences of blocking VAMP1/2/3-dependent exocytosis from oligodendrocyte-lineage cells, we assessed the motor behavior of PD:ibot and littermate control mice using the rotarod test. We placed mice on a gradually accelerating rotarod and recorded the time each mouse stayed on the rotarod. We found that PD:ibot mice stayed on the rotarod for significantly shorter amounts of time than littermate control mice on all 3 days of testing (*Figure 3P-S*). Therefore, blocking VAMP1/2/3-dependent exocytosis from oligodendrocyte-lineage cells led to deficits in neural circuit function.

## PD:ibot mice exhibit changes in the transcriptomes of OPCs and oligodendrocytes

We next aimed to uncover the molecular changes in OPCs and oligodendrocytes in PD:ibot mice. We performed immunopanning to purify OPCs and oligodendrocytes from the brains of P17 PD:ibot and littermate control mice and performed RNA-sequencing (RNA-seq). We detected broad and robust gene expression changes in oligodendrocytes, and moderate changes in OPCs (*Figure 4A and B*, *Supplementary files 1 and 2*), demonstrating that VAMP1/2/3-dependent exocytosis from oligodendrocyte-lineage cells is critical for establishing and/or maintaining the normal molecular attributes of oligodendrocytes and OPCs. Notably, the expression of signature genes of differentiated oligodendrocytes such as *Plp1, Mbp, Aspa, and Mobp* was significantly reduced in oligodendrocytes purified from PD:ibot mice compared with controls (*Figure 4E–H*, *Supplementary files 1 and 2*). This result was not secondary to a reduction in oligodendrocyte density, as we loaded a similar amount of cDNA libraries from PD:ibot and control oligodendrocytes for sequencing, and processed all sequencing data with the same pipeline. Therefore, VAMP1/2/3-dependent exocytosis from oligodendrocyte-lineage cells is critical for the expression of mature oligodendrocyte genes. The expression of genes encoding oligodendrocyte-lineage cell marker proteins used for immunohistochemistry, *Qk* (encoding the Qk protein recognized by the CC1 antibody), *Olig2*, and PDGFRα did not change in PD:ibot mice (*Figure 4—figure supplement 1*). Therefore, our oligodendrocyte count is not confounded by changes in marker gene expression. We next performed gene ontology (GO) analysis to reveal the molecular pathways and cellular processes altered in each type of glial cell in PD:ibot mice (*Figure 4I–L*, *Supplementary file 3*). Genes associated with filopodium assembly, calcium ion transport, and plasma membrane raft assembly pathways were increased, whereas genes associated with lipid biosynthetic process, axon ensheathment, and myelination pathways were reduced in oligodendrocytes in PD:ibot mice. Genes associated with the trans-synaptic signaling, chemical synaptic transmission, and GPCR signaling pathways were increased in OPCs in PD:ibot mice. To assess whether oligodendrocyte-lineage cell exocytosis affects other glial cell types, such as astrocytes and microglia, we also purified these cells by immunopanning and performed RNA-seq. We observed moderate

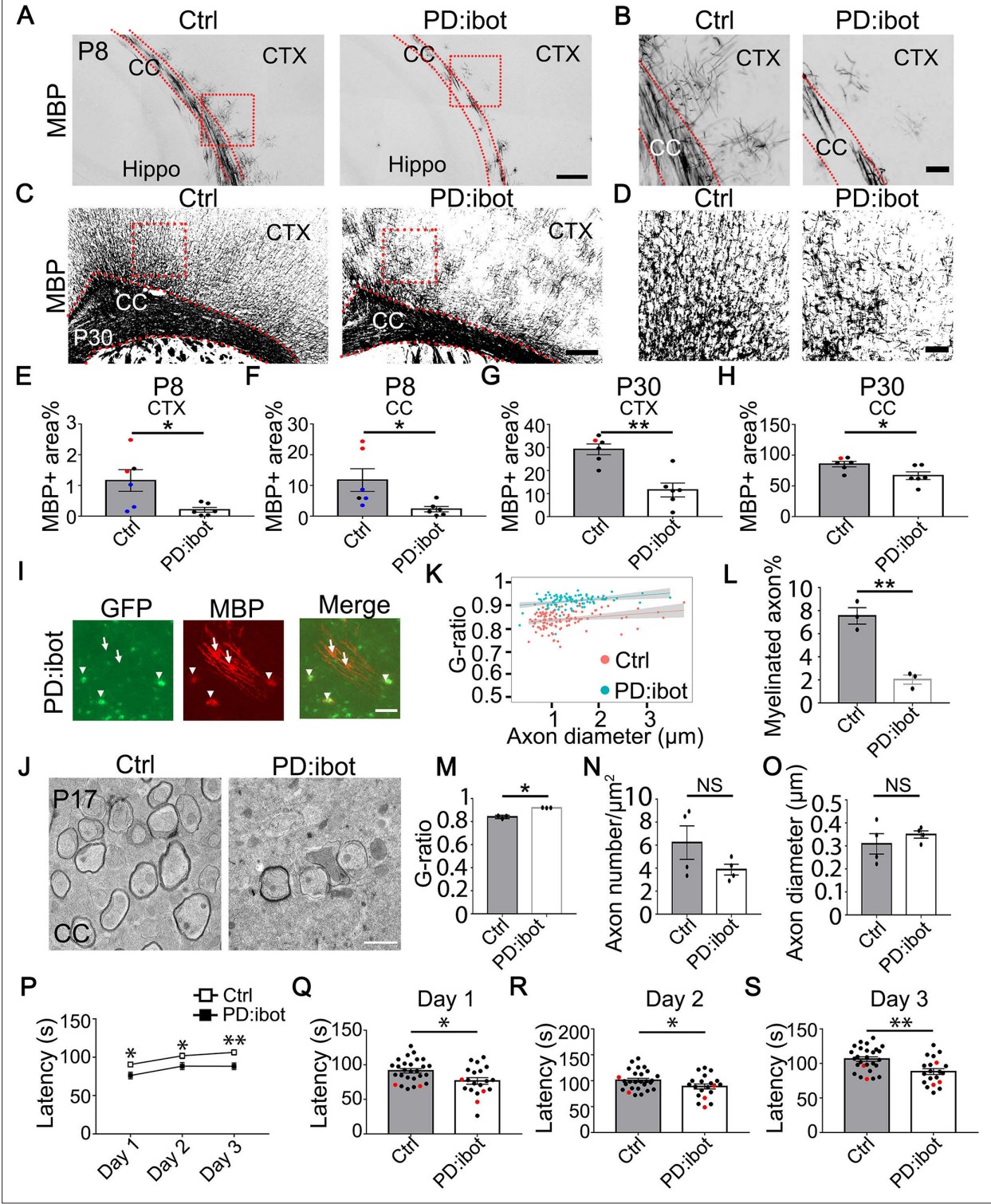

**Figure 3.** Defective myelination and motor behavior in PD:ibot mice. (**A–D**) MBP immunofluorescence at P8 (**A, B**) and P30 (**C, D**) in PD:ibot and control brains. Dashed lines delineate the corpus callosum. CTX, cerebral cortex. CC, corpus callosum. Hippo, hippocampus. Boxed areas in (**A**) are enlarged and shown in (**B**); Boxed areas in (**C**) are enlarged and shown in (**D**). Scale bars: 200 μm in (**A, C**), 50 μm in (**B, D**). (**E–H**) Quantification of MBP +area in the cerebral cortex (**E, G**) and the corpus callosum (**F, H**) at P8 and P30. Control mice at P8 are ibot only (black dots), Cre only (red dot), and wildtype (blue dots). Control mice at P30 are ibot only (black dots) and Cre only (red dots). N=6 mice per group. Paired two-tailed T-test. MBP coverage in the cortex (%): 1.16±0.35 in control and 0.21±0.079 in PD:ibot, p=0.030 at P8; 29.21±2.22 in control and 11.55±3.02 in PD:ibot, p=0.0025 at p30. MBP coverage in the corpus callosum (%): 11.73±3.70 in control and 2.26±0.87 in PD:ibot, p=0.024 at P8; 85.50±4.40 in control and 66.61±6.25 in PD:ibot,

*Figure 3 continued on next page*

*Figure 3 continued*

p=0.024 at P30. (**I**) The morphology of ibot-GFP + cells and GFP⁻ control cells labeled by MBP immunofluorescence. A region in the cerebral cortex from a P8 PD:ibot mouse is shown. The arrowheads point to ibot-GFP + cells and the arrows point to GFP⁻ control cells. Scale bar: 50 μm. (**J**) Transmission electron microscopy images of the corpus callosum at P17. Scale bar: 1 μm. (**K**) *g-ratio* as a function of axon diameter in the corpus callosum at P17. N=3 mice per group. 112 myelinated axons from control and 97 myelinated axons from PD:ibot were analyzed. (**L, M**) Quantification of the percentage of myelinated axons and *g-ratio* (axon diameter divided by the diameter of myelin +axon) from the transmission electron microscopy images of the corpus callosum at P17. N=3 mice per group. All control mice at P17 are ibot only (black dots). Paired two-tailed T-test. Myelinated axons %: 7.6±0.7 in control and 2.0±0.4 in PD:ibot, p=0.0048. *g-ratio*: 0.84±0.009 in control and 0.92±0.0009 in PD:ibot; p=0.012. (**N, O**) Quantification of axon density and axon diameter from the transmission electron microscopy images of the corpus callosum at P17. N=4 mice per group. Paired two-tailed T-test. Axon number/μm²: 6.2±1.5 in control and 3.9±0.47 in PD:ibot, p=0.19. Axon diameter: 309.1±44.3 nm in control and 349.8±16.0 nm in PD:ibot; p=0.51. (**P**) Latency to fall from an accelerating rotarod (seconds). Each mouse was tested three times per day for 3 consecutive days. The average latency to fall of the three trials of each mouse was recorded for each day. No significant sex differences were detected. Unpaired two-tailed T-test was performed with Benjamini, Krieger, and Yekutieli's false discovery rate (FDR) method to correct for multiple comparisons. *, FDR <0.05. **, FDR <0.01. (**Q–S**) Latency to fall on each testing day. Day 1: PD:ibot: 76.7±4.7 seconds, control: 91.4±3.2 s, p=0.015; day 2: PD:ibot: 88.9±4.9 s, control: 101.1±3.7 s, p=0.033; day 3: PD:ibot: 88.4±4.3 s, control: 106.4±3.3 s, p=0.0051. N=27 mice for control and 20 mice for PD:ibot. 2 months old mice (black dots) and 5 months old mice (red dots) are pooled for experiments.

The online version of this article includes the following source data and figure supplement(s) for figure 3:

**Source data 1.** It contains original data points for *Figure 3E, F, G, H, K, L, M, N, O, P, Q, R and S*.

**Figure supplement 1.** No change in the percentage of cleaved-caspase-3⁺ cells in oligodendrocytes, OPCs, and other lineages.

**Figure supplement 1—source data 1.** It contains original data points for *Figure 3—figure supplement 1B, C, D and E*.

**Figure supplement 2.** Ibot-GFP-expressing cells contribute to the population of surviving differentiated oligodendrocytes.

**Figure supplement 2—source data 1.** It contains original data points for *Figure 3—figure supplement 2A*.

**Figure supplement 3.** Increase in the percentage of Ki-67⁺ proliferating OPCs in PD:ibot mice at P8.

**Figure supplement 3—source data 1.** It contains original data points for *Figure 3—figure supplement 3B*.

changes in astrocytes and microglia (*Figure 4C and D*). For example, genes associated with phagocytosis, such as *Cd68* and *C1qc*, were increased in microglia from PD:ibot mice (*Supplementary file 2*), suggesting the importance of oligodendrocyte exocytosis in oligodendrocyte-microglial interactions.

## Blocking VAMP1/2/3-dependent exocytosis from oligodendrocyte-lineage cells impairs oligodendrocyte development in vitro

VAMP1/2/3-dependent exocytosis from oligodendrocyte-lineage cells may directly affect oligodendrocyte development or change the attributes of other cell types, and, in turn, indirectly affect oligodendrocytes. For example, OPC-secreted molecules may affect axonal growth, and subsequently, axonal signals may affect oligodendrocytes indirectly. Therefore, we next employed purified OPC and oligodendrocyte cultures to determine whether exocytosis has direct roles in oligodendrocyte-lineage cells in the absence of other cell types.

We performed immunopanning to purify OPCs from P7 PD:ibot and control mice injected with 4-hydroxytamoxifen as described above. We cultured the OPCs for two days in the proliferation medium and then switched to the differentiation medium and cultured them for another seven days. To assess oligodendrocyte differentiation and maturation, we assessed the levels of MBP protein, a marker for differentiated oligodendrocytes, and detected lower MBP levels in PD:ibot cells compared with controls (*Figure 5B and C*). Similarly, *Mbp* mRNA levels were also lower as determined by quantitative real-time PCR (*Figure 5D*).

Additionally, we assessed the morphological maturation of oligodendrocytes in vitro (*Figure 5E–H*). OPCs are initially bipolar, and as they differentiate, they grow a few branches to become star-like. The cells next grow more branches to become arborized and then extend myelin-sheath-like flat membranous structures, acquiring a 'lamellar' morphology (*Figure 5A*; *Zuchero et al., 2015*) (also referred to as a 'fried egg' or 'pancake' morphology). We used the CellMask dye to analyze the morphological maturation of oligodendrocytes. At day 3 of differentiation, we found that a larger proportion of PD:ibot cells than control cells are at the early 'star', stage whereas a smaller proportion of PD:ibot cells than control cells have proceeded to the late 'lamellar' stage (*Figure 5F*). At day 7 of differentiation, more PD:ibot cells have proceeded from the 'star' to the 'arborized' stage compared with day 3, but the percentage of cells that have proceeded to the late 'lamellar' stage remains lower in PD:ibot

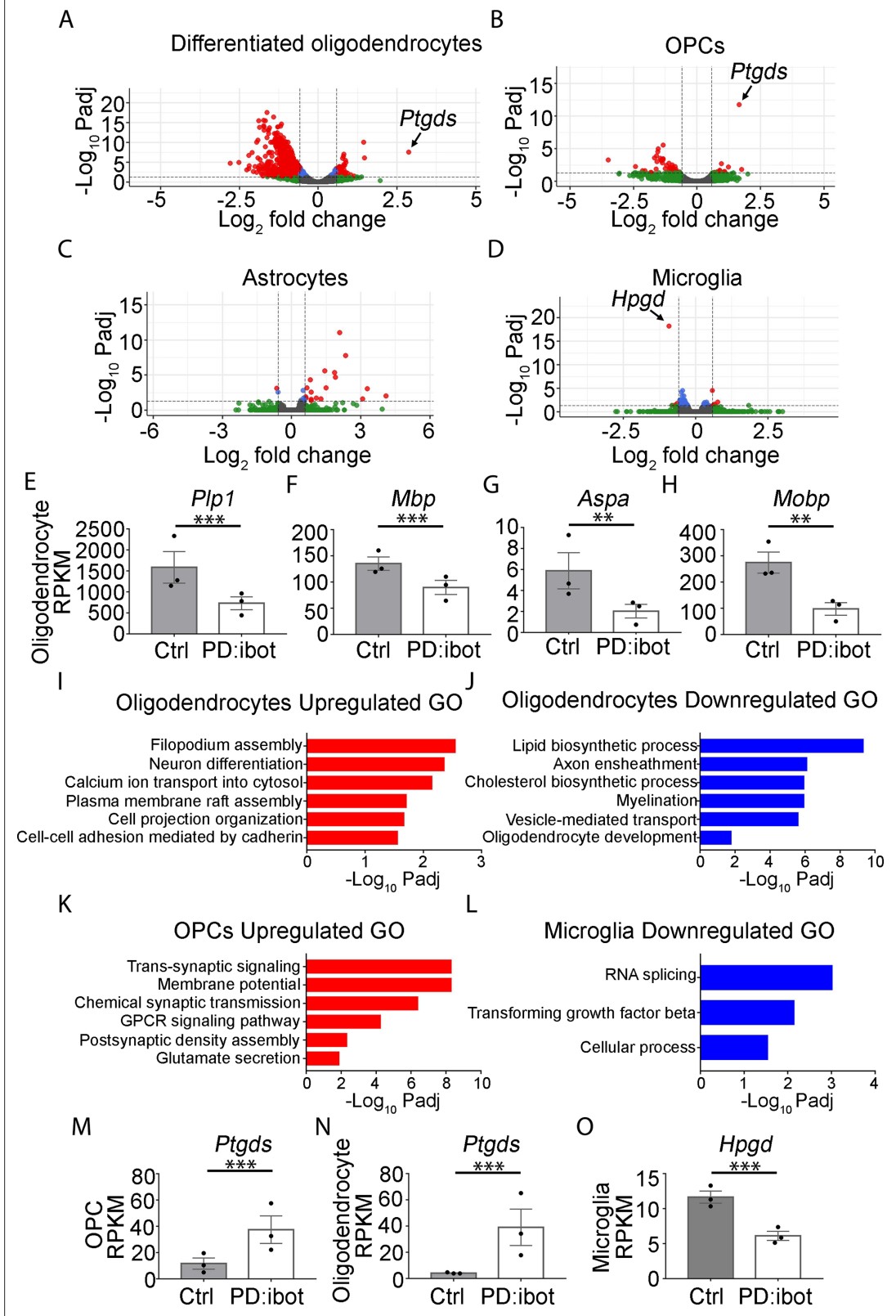

**Figure 4.** Transcriptome changes of purified glial cells from PD:ibot mice. (**A–D**) Differentiated oligodendrocytes, OPCs, astrocytes, and microglia were purified by immunopanning from whole brains of P17 PD:ibot and littermate control mice. Anti-GalC hybridoma was used to purify differentiated oligodendrocytes; anti-O4 hybridoma was used to isolate OPCs; anti-HepaCAM antibody was used to purify astrocytes; anti-CD45 antibody was used to isolate microglia. More details can be found in the method section. Gene expression was determined by RNA-seq. Genes exhibiting significant

*Figure 4 continued on next page*

*Figure 4 continued*

changes (P-value adjusted for multiple comparisons, Padj <0.05, fold change >1.5) are shown in red. (**E–H**) Examples of mature oligodendrocyte marker gene expression by oligodendrocytes purified from PD:ibot and control mice at P17. N=3 mice per group. Significance is determined by DESeq2. *Plp1*: 1586±375.9 in control and 729.3±152.4 in PD:ibot, p=3.96 × 10$^{-13}$; *Mbp*: 135.2±12.8 in control and 89.8±13.5 in PD:ibot, p=4.76 × 10$^{-5}$; *Aspa*: 5.9±1.7 in control and 2.0±0.6 in PD:ibot, p=0.003; *Mobp*: 274.4±40.1 in control and 97.5±24.0 in PD:ibot, p=0.0004. (**I–L**) Examples of GO terms associated with genes upregulated in oligodendrocytes (**I**), downregulated in oligodendrocytes (**J**), upregulated in OPCs (**K**), and downregulated in microglia (**L**). There are no GO terms significantly associated with genes downregulated in OPCs or upregulated in microglia. (**M, N**) Expression of *Ptgds* by OPCs and oligodendrocytes purified from PD:ibot and control mice at P17. N=3 mice per group. Significance is determined by DESeq2. *Ptgds* in OPCs: 11.6±4.3 in control and 34.5±10.5 in PD:ibot, p=1.79 × 10$^{-12}$; *Ptgds* in oligodendrocytes: 4.1±0.3 in control and 39.1±13.9 in PD:ibot, p=3.14 × 10$^{-8}$. (**O**) Expression of *Hpgd* by microglia purified from PD:ibot and control mice at P17. (**E–H, M–O**) Expression is shown in RPKM. N=3 mice per group. Significance is determined by DESeq2. *Hpgd* in microglia: 11.7±0.9 in control and 6.1±0.7 in PD:ibot, p=6.34 × 10$^{-19}$.

The online version of this article includes the following source data and figure supplement(s) for figure 4:

**Source data 1.** It contains original data points for *Figure 4E, F, G, H, M, N and O*.

**Figure supplement 1.** The expression of *Qk*, *Olig2*, and *Pdgfra* did not change in the oligodendrocyte-lineage cells in PD:ibot mice at P17 (**A–C**) Expression of *Qk*, *Olig2*, and *Pdgfra* by oligodendrocytes purified from PD:ibot and control mice.

**Figure supplement 1—source data 1.** It contains original data points for *Figure 4—figure supplement 1A,B,C,D,E and F*.

than in control cultures (*Figure 5E and G*). We next quantified the size of lamellar cells, which have large sheaths of myelin-like membrane. Interestingly, we found that lamellar cells from PD:ibot mice are significantly smaller than those from control mice (*Figure 5E and H*). We also used a membrane-staining version of the CellMask dye at day 7 of differentiation and found similar results (*Figure 5—figure supplement 1*). Together, these observations suggest that VAMP1/2/3-dependent exocytosis is required for the morphological maturation of oligodendrocytes and that exocytosis has a direct effect on cells within the oligodendrocyte lineage to promote their development.

## Cell non-autonomous effect of botulinum toxin-B in oligodendrocyte development in vitro

To examine whether cell non-autonomous effect contributes to the oligodendrocyte development defect associated with botulinum toxin-B expression, we compared the growth of wild-type cells in cultures containing *vs.* not containing botulinum-expressing cells. We took advantage of the fact that all OPCs purified from PD:ibot mice were not botulinum-GFP-expressing (efficiency ~65%, *Figure 6A and B*). The GFP$^-$ cells in PD:ibot OPC cultures did not express botulinum toxin and were competent in exocytosis. We compared the development of GFP$^-$ control cells in cultures generated from PD:ibot mice *vs.* control cells in cultures generated from control mice. Interestingly, we found that the percentages and the sizes of lamellar cells in control cells from PD:ibot cultures were smaller than in control cells from control cultures (*Figure 6C and D*). Although both groups of cells were competent in exocytosis, they were surrounded by exocytosis-deficient *vs.* exocytosis-competent neighbor cells. The difference in the growth capacity of control cells in the presence of different neighbor cells reveals cell non-autonomous contributions of botulinum-expressing cells in oligodendrocyte development.

## Oligodendrocyte-lineage cell-secreted molecules partially restore oligodendrocyte morphological maturation in secretion-deficient cells

Cell non-autonomous effects on oligodendrocyte development may be mediated by contact-dependent mechanisms or secreted molecules. To distinguish between these possibilities, we next assessed whether adding oligodendrocyte-lineage cell-secreted molecules could restore differentiation in VAMP1/2/3-dependent exocytosis-deficient OPCs. We prepared co-cultures of OPCs separated by inserts with 1 μm-diameter pores to allow for the diffusion of secreted molecules (*Figure 6E*). We plated PD:ibot and control cells on inserts and on the bottom of culture wells in four combinations: (1) control-inserts-control-wells; (2) PD:ibot-inserts-control-wells; (3) PD:ibot-inserts-PD:ibot-wells; and (4) control-inserts-PD:ibot-wells (*Figure 6F*) and examined oligodendrocyte morphological differentiation on the bottom of culture wells by quantifying lamellar cells as described above. We performed one-way ANOVA with a multiple comparison test comparing every group to every other group. Comparing group 3 vs. group 4, we found that adding secreted molecules from control cells on inserts partially rescued the size of lamellar cells of PD:ibot cells on the bottom of culture wells

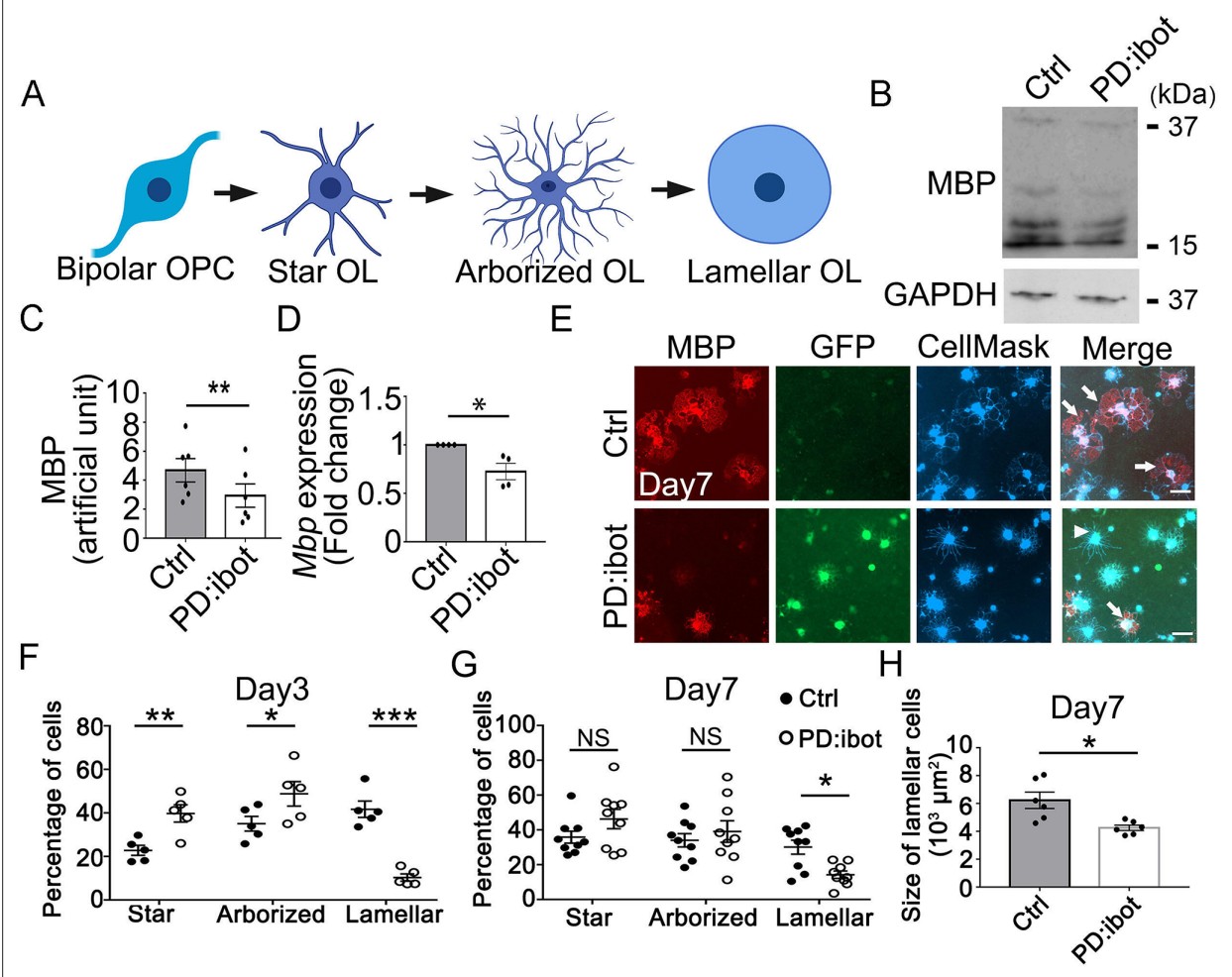

**Figure 5.** In vitro development defect of oligodendrocytes purified from PD:ibot mice. (**A**) A diagram of the morphological changes during oligodendrocyte differentiation in vitro. OPCs exhibit a bipolar morphology. Differentiating oligodendrocytes first grow multiple branches (star-shaped and arborized) and then develop myelin-like membrane extension and exhibit a lamellar morphology. (**B**) Western blot for MBP in oligodendrocyte cultures after 7 days of differentiation. (**C**) Quantification of MBP proteins from Western blot. N=6 mice per group. Paired two-tailed T-test. MBP intensity: 4.7±0.8 in control and 2.9±0.8 in PD:ibot, p=0.001. (**D**) Quantification of *Mbp* mRNAs based on qPCR. N=4 mice per group. Paired two-tailed T-test. *Mbp* relative expression: 1±0.0 in control and 0.7±0.08 in PD:ibot, p=0.05. (**E**) Oligodendrocyte cultures after 7 days of differentiation. Red, MBP. Green, ibot:GFP. Blue, CellMask, which labels all cells. Arrows point to examples of lamellar cells and an arrowhead points to an example of a star-shaped oligodendrocyte. Scale bars: 50 μm. (**F–G**) Quantification of the percentage of cells at the star-shaped, arborized, and lamellar stages after 3 (**F**) and 7 (**G**) days of differentiation. Filled circles, control. Open circles, PD:ibot. N=5 mice per group on day 3 and N=9 mice per group on day 7. Multiple T-test with two-stage set-up method of Benjamini, Krieger, and Yekutieli for multiple comparisons. Star stage at day 3 (%): 22.8±2.3 in control and 39.8±4.0 in PD:ibot, p=0.003; arborized stage at day 3 (%): 35.1±3.4 in control and 48.8±5.6 in PD:ibot, p=0.014; lamellar stage at day 3 (%): 41.7±3.7 in control and 10.3±1.7 in PD:ibot, p=0.3 × $10^{-5}$. Star stage at day 7 (%): 35.9±3.5 in control and 46.3±5.6 in PD:ibot, p=0.1; arborized stage at day 7 (%): 34.0±3.8 in control and 39.1±6.2 in PD:ibot, p=0.4; lamellar stage at day 7 (%): 30.1±4.0 in control and 14.2±2.1 in PD:ibot, p=0.01. (**H**) Quantification of the size of lamellar cells in oligodendrocyte cultures obtained from PD:ibot and littermate control mice after 7 days of differentiation. N=6 cultures from 4 mice per group. Paired two-tailed T-test. 6237±587.5 μm$^2$ in control and 4253±193.7 μm$^2$ in PD:ibot; p=0.016.

The online version of this article includes the following source data and figure supplement(s) for figure 5:

**Source data 1.** It contains original data points for *Figure 5C, D, G, F, G and H*.

**Source data 2.** It contains figures with the uncropped blots with the relevant bands clearly labelled (**A**) and original blot images (**B and C**).

**Figure supplement 1.** In vitro development defect of oligodendrocytes purified from PD:ibot mice (**A**) Oligodendrocyte cultures after 7 days of differentiation.

**Figure supplement 1—source data 1.** It contains original data points for *Figure 5—figure supplement 1B and C*.

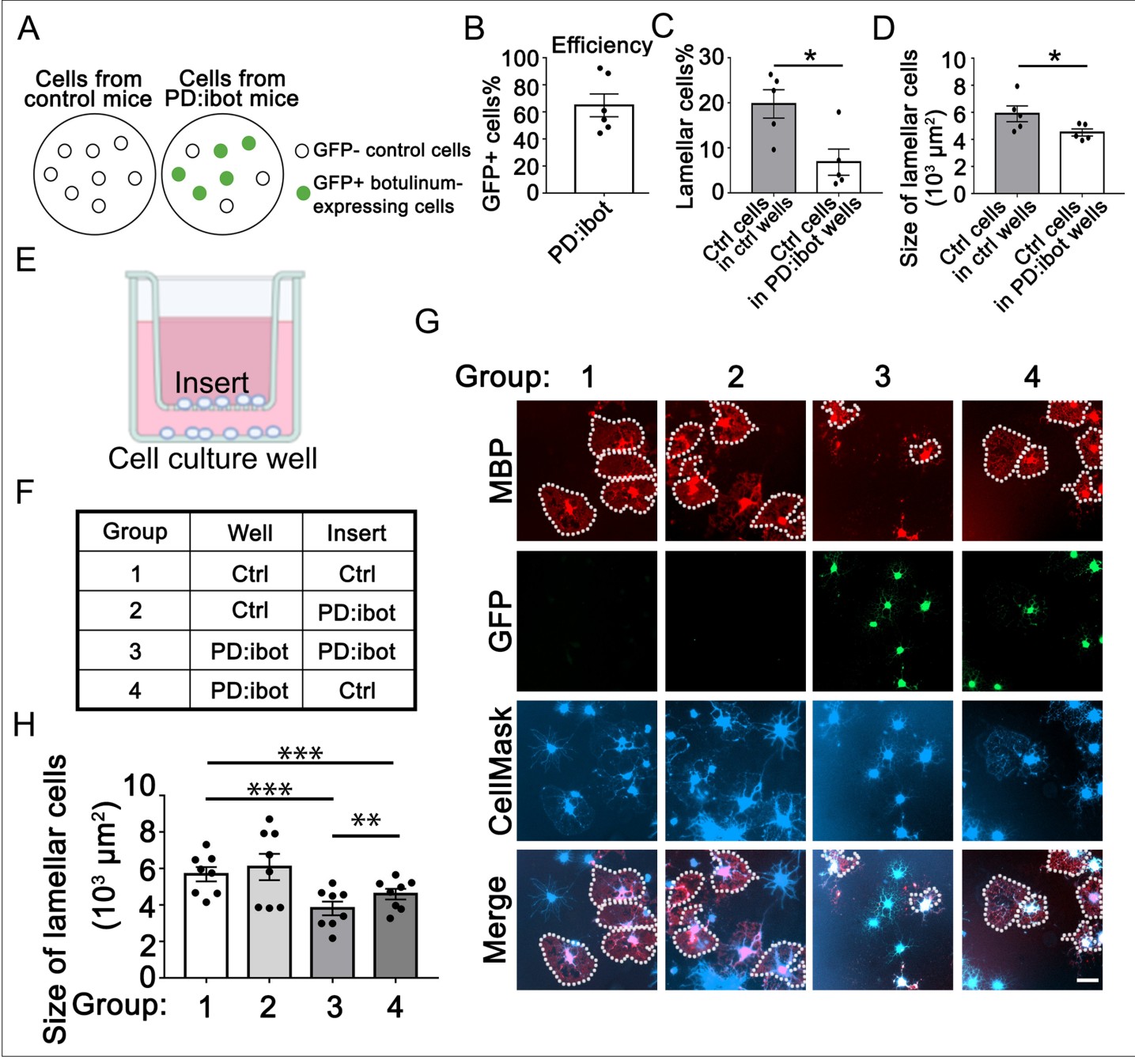

**Figure 6.** Cell non-autonomous effect of botulinum toxin-B in oligodendrocyte development in vitro. (**A**) A diagram of GFP⁻ control cells and GFP +botulinum-expressing cells in control culture and PD:ibot culture. (**B**) Quantification of the percentage of GFP + cells in PD:ibot cultures. N=6 mice. (**C**) Quantification of the percentage of lamellar cells after 7 days of differentiation. N=5 mice per group. Paired two-tailed T-test. Lamellar cell percentage (%): 19.76±3.16 in control and 6.82±2.92 in PD:ibot; p=0.023. (**D**) Quantification of the size of lamellar cells after 7 days of differentiation. N=5 mice per group. Paired two-tailed T-test. 5897±585.4 µm² in control and 4520±261.6 µm² in PD:ibot; p=0.027. (**E**) A diagram of the cocultures of cells separated by a porous insert with 1 µm pore size. (**F**) The genotype of cells on the inserts and wells in each group. The differentiation of the cells on the bottom of the wells was examined. (**G**) Oligodendrocyte cultures after 7 days of differentiation. Red, MBP. Green, ibot:GFP. Blue, CellMask, which labels all cells. Dotted lines delineate examples of lamellar cells. Scale bars: 50 µm. (**H**) Quantification of the size of lamellar cells. N=8 cultures from 5 mice per condition. Both GFP⁺ and GFP⁻ cells were included in the quantification. One-way ANOVA with Benjamini, Krieger, and Yekutieli's two-stage linear step-up FDR procedure for multiple comparisons. Every group was compared with every other group. Group 3 vs. group 4: p=0.0010. Group 1 vs. group 3: p=<0.0001. Group 2 vs. group 3: p=0.0006. Group 1 vs. group 4: p=0.0006. Group 2 vs. group 4: p=0.0055. Group 1 vs. group 2: p=0.064. Size of lamellar cells: group 1: 5691±391 µm²; group 2: 6087±720.7 µm²; group 3: 3810±376 µm²; group 4: 4594±293.3 µm².

The online version of this article includes the following source data and figure supplement(s) for figure 6:

*Figure 6 continued on next page*

*Figure 6 continued*

**Source data 1.** It contains original data points for *Figure 6B, C, D and H*.

**Source data 2.** It contains raw data for table in *Figure 6F*.

**Figure supplement 1.** Cell non-autonomous effect of botulinum toxin-B in oligodendrocyte development in vitro (**A**) A diagram of the cocultures of cells separated by a porous insert with 1 μm pore size.

**Figure supplement 1—source data 1.** It contains original data points for *Figure 6—figure supplement 1D*.

**Figure supplement 1—source data 2.** It contains raw data for table in *Figure 6—figure supplement 1B*.

(*Figure 6G and H*). We used both a cytosol-staining version (*Figure 6G*) and a membrane-staining version (*Figure 6—figure supplement 1*) of the CellMask dye and found similar results. These observations lend further support to the hypothesis that oligodendrocyte-lineage cell-secreted molecules promote oligodendrocyte development.

## Blocking L-PGDS leads to oligodendrocyte development defects in vitro

We next sought to uncover the identity of the secreted molecules that promote oligodendrocyte and myelin development. We mined our oligodendrocyte and OPC RNA-seq dataset to identify highly expressed genes encoding secreted proteins (*Zhang et al., 2014*; *Zhang et al., 2016*). After testing a few candidate genes using in vitro OPC cultures, we focused on *Ptgds*, one of the most abundant genes encoding a secreted protein expressed by oligodendrocyte-lineage cells in both humans and mice (*Zhang et al., 2014*; *Zhang et al., 2016*). Its expression increases during development (*Kang et al., 2011*) as oligodendrocyte development and myelination occur. Interestingly, *Ptgds* is important for Schwann cell myelination in the peripheral nervous system (*Trimarco et al., 2014*). Yet, its function in the development of the CNS is unknown. *Ptgds* encodes the L-PGDS protein, which converts prostaglandin H2 to prostaglandin D2 (PGD2) (*Urade and Hayaishi, 2000*). We performed western blot analyses to assess L-PGDS secretion by botulinum toxin B-expressing OPCs/oligodendrocytes in culture. We detected an increase in intracellular L- PGDS in OPC/oligodendrocyte cultures from PD:ibot mice compared with controls (*Figure 7A*). Secreted L-PGDS, however, is lower in PD:ibot compared with control cultures (*Figure 7A–C*), suggesting that botulinum toxin B inhibits L-PGDS secretion. L-PGDS secretion is not completely eliminated, most likely because not all cells in the culture express botulinum toxin (efficiency:~65%, *Figure 6B*) and wild-type cells may compensate by increasing secretion when extracellular L-PGDS levels are low.

To determine the role of L-PGDS in oligodendrocyte development and CNS myelination, we first assessed oligodendrocyte development in vitro in the presence of AT-56, a specific L-PGDS inhibitor (*Irikura et al., 2009*). We found that AT-56 inhibits wild-type oligodendrocyte development in a dose-dependent manner in vitro (*Figure 7D and E*), suggesting a requirement of L-PGDS in oligodendrocyte development, without affecting their survival (*Figure 7—figure supplement 1*). Using cytoplasm-staining and membrane-staining versions of CellMask dyes, we observed similar results (*Figure 7—figure supplement 2*).

L-PGDS synthesizes PGD2, whereas another enzyme, 15-hydroxyprostaglandin dehydrogenase (HPGD) inactivates PGD2 (*Conner et al., 2001*). To assess the involvement of PGD2 in oligodendrocyte development, we added HPGD to OPC cultures from control mice. Interestingly, we found that HPGD reduced the percentage of oligodendrocytes with mature lamellar morphology (*Figure 7F and G*), lending further support to the potential role of the L-PGDS/PGD2 pathway in oligodendrocyte development.

## L-PGDS is required for oligodendrocyte development in vivo

Having discovered the role of L-PGDS in oligodendrocyte development in vitro, we next assessed whether L-PGDS regulates oligodendrocyte development in vivo. We examined oligodendrocytes in L-PGDS global knockout mice and found a significant decrease in CC1[+] oligodendrocytes in the corpus callosum and cerebral cortex of L-PGDS-knockout mice at P9 (*Figure 8A and B*). The density of OPCs (PDGFRα[+]) and cleaved caspase-3 positive cells and the intensity of SMI-32, a marker for damaged axons, and Iba1, a marker for microglial reactivity, did not differ between L-PGDS-knockout and control mice (*Figure 8C–K*).

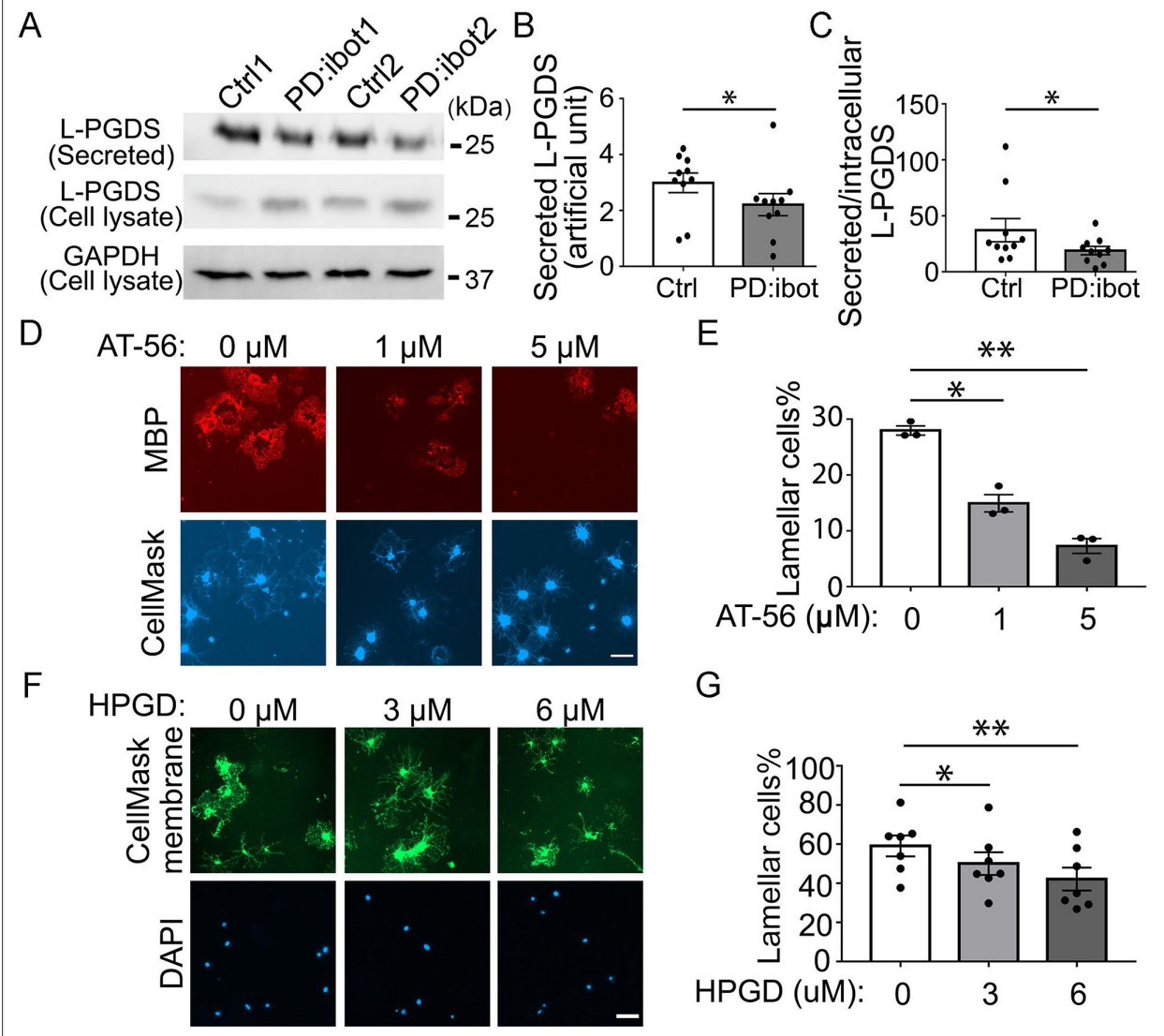

**Figure 7.** The effect of the L-PGDS inhibitor, AT-56, and HPGD, which inactivates PGD2, on oligodendrocyte development in vitro. (**A**) Immunoblot of secreted and intracellular L-PGDS protein from oligodendrocyte cultures from PD:ibot and littermate control mice. (**B**) Quantification of the immunoblot signal intensity of secreted L-PGDS. N=10 mice per group. Paired two-tailed T-test. Secreted L-PGDS intensity: 3.0±0.4 in control and 2.2±0.4 in PD:ibot, p=0.0104. (**C**) Quantification of the ratio of secreted and intracellular L-PGDS. N=10 mice per group. Paired two-tailed T-test. Ratio of secreted and intracellular L-PGDS: 37.2±10.4 in control and 18.9±3.7 in PD:ibot, p=0.05. (**D**) Oligodendrocyte cultures from wild-type mice after 7 days of differentiation in the presence and absence of the L-PGDS inhibitor AT-56. Red: MBP immunofluorescence. Blue: CellMask, which labels all cells. Scale bars: 50 µm. (**E**) Quantification of cells with lamellar morphology. One-way ANOVA with Benjamini, Krieger, and Yekutieli's two-stage linear step-up FDR procedure for multiple comparisons. N=3 cultures from 3 mice per group. Lamellar cells%: DMSO control: 28±0.8; 1 µM AT-56: 15±1.6, p=0.024; 5 µM AT-56: 7.3±1.3, p=0.0068. (**F**) Oligodendrocyte cultures from wild-type mice after 7 days of differentiation in the presence and absence of HPGD. Green: Membrane version of CellMask, which labels the membranes of all cells. Blue: DAPI. Scale bars: 50 µm. (**G**) Quantification of cells with lamellar morphology. One-way ANOVA with Benjamini, Krieger, and Yekutieli's two-stage linear step-up FDR procedure for multiple comparisons. N=7 cultures from 6 mice per group. Lamellar cells%: 0 µM HPGD: 59.10±5.33; 3 µM HPGD: 50.09±5.80, p=0.034; 6 µM HPGD: 42.20±5.87, p=0.0090.

The online version of this article includes the following source data and figure supplement(s) for figure 7:

**Source data 1.** It contains original data points for *Figure 7B, C, E and G*.

**Source data 2.** It contains figures with the uncropped blots with the relevant bands clearly labelled (**A**) and original blot images (**B and C**).

**Figure supplement 1.** No change in the percentage of apoptotic cells in AT-56 treatment.

**Figure supplement 1—source data 1.** It contains original data points for *Figure 7—figure supplement 1B*.

**Figure supplement 2.** AT-56 inhibit oligodendrocytes development in vitro.

**Figure supplement 2—source data 1.** It contains original data points for *Figure 7—figure supplement 2B*.

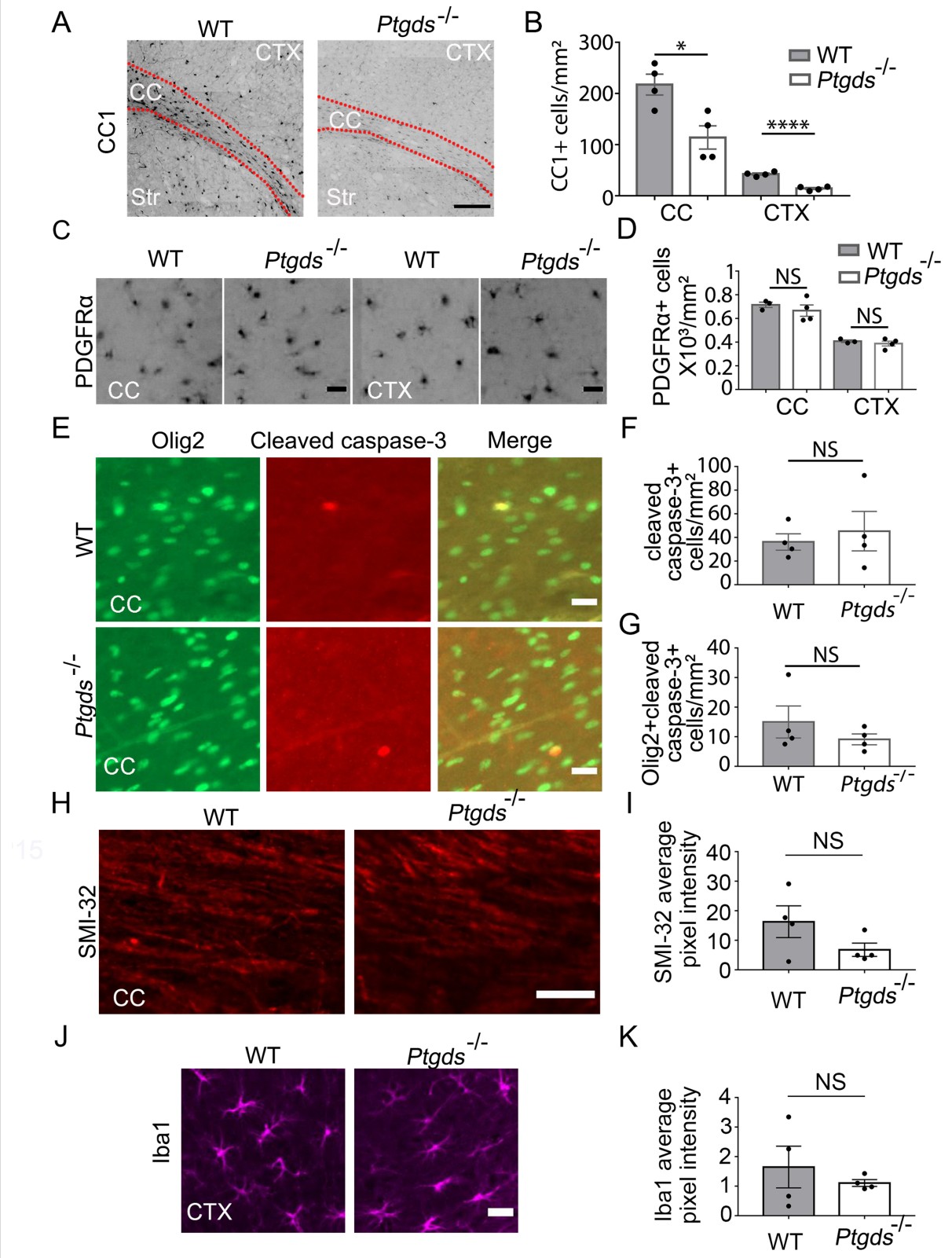

**Figure 8.** Oligodendrocyte defect in L-PGDS-knockout mice. (**A**) CC1 immunofluorescence at P9. The dashed lines delineate the corpus callosum (CC). Ctx, cortex. Str, striatum. Scale bar: 200 μm. (**B**) Quantification of the density of CC1$^+$ cells. N=4 mice per genotype. Corpus callosum: 217.6±20.3 / mm$^2$ in control, 114±22.8 in knockout, p=0.015; cerebral cortex: 42.5±2.2 in control, 14.3±2.0 in knockout, p=0.0001. Unpaired two-tailed T-test in all quantifications in this figure. (**C**) PDGFRα immunofluorescence at P9 in the corpus callosum and the cerebral cortex. Scale bar: 20 μm. (**D**) Quantification

*Figure 8 continued on next page*

*Figure 8 continued*

of the density of PDGFRα⁺ cells. Corpus callosum: 715.3±22.7 in control, 667±48.0 in knockout, p=0.46; cerebral cortex: 406.2±10.3 in control, 388.3±20.8 in knockout, p=0.48. (**E**) Cleaved caspase-3 immunofluorescence in L-PGDS-knockout and littermate control mice. Scale bar: 20 µm. (**F**) Quantification of the density of cleaved caspase-3⁺ cells in the corpus callosum. 36.2±6.9 in control, 45.3±16.7 in knockout, p=0.63. (**G**) Quantification of the density of cleaved caspase-3⁺ oligodendrocyte-lineage cells in corpus callosum. 15.0±5.4 in control, 9.1±1.8 in knockout, p=0.34. (**H**) SMI-32 immunofluorescence at P9 in the corpus callosum. Scale bar: 50 µm. (**I**) Quantification of the average pixel intensity of SMI-32 in the corpus callosum. Unpaired two-tailed T-test. N=4 mice per genotype. 16.3±5.4 in control, 6.8±2.3 in knockout, p=0.16. (**J**) Iba1 immunofluorescence at P9 in the cortex. Scale bar: 20 µm. (**K**) Quantification of the average pixel intensity of Iba1 in the cortex. Unpaired two-tailed T-test. N=4 mice per genotype. 1.6±0.71 in control, 1.1±0.12 in knockout, p=0.48.

The online version of this article includes the following source data for figure 8:

**Source data 1.** It contains original data points for *Figure 8B, D, F, G, I and K*.

## L-PGDS and PGD2 restore the development of secretion-deficient oligodendrocytes

Next, we determined whether L-PGDS is sufficient to rescue the maturation defect of PD:ibot cells in vitro. Indeed, the exogenous addition of recombinant L-PGDS protein partially rescued the percentage of cells with lamellar morphology from PD:ibot mice (*Figure 9A and B*), further supporting the role of L-PGDS in oligodendrocyte development. To assess the role of PGD2, the synthesis product of the L-PGDS enzyme, in oligodendrocyte development, we added PGD2 to cultures from PD:ibot mice and observed a partial rescue of the oligodendrocyte maturation defect of PD:ibot cells (*Figure 9C and D*). Although other cell types could mediate the effects of systemic L-PGDS knockout, the observation that L-PGDS and PGD2 rescue the morphological maturation of oligodendrocytes in purified cultures in vitro supports a direct role of PGD2 in oligodendrocyte-lineage cell development.

## Overexpression of L-PGDS partially rescues the myelin defect in PD:ibot mice in vivo

To assess the effect of L-PGDS on myelin development in vivo, we bred the L-PGDS overexpressing transgenic mouse strain (*Ptgds*-TG, line B7) (*Pinzar et al., 2000*) with PD:ibot mice to obtain triple transgenic mice (*Pdgfra-CreER*; lox-stop-lox-botulinum toxin-B light chain-GFP, *Ptgds*-TG). Compared with PD:ibot mice, the triple transgenic mice exhibited enhanced myelination (*Figure 9E–H*), providing further evidence of the role of L-PGDS in oligodendrocyte and myelin development in vivo. Interestingly, we found that *Ptgds* is the most highly upregulated gene in OPCs and oligodendrocytes from PD:ibot mice compared with control mice based on RNA-seq (*Figure 4A and B*). Moreover, *Hpgd*, which inactivates PGD2, is the most robustly downregulated gene in microglia from PD:ibot mice compared with control mice (*Figure 4D*). These intriguing observations may reflect homeostatic mechanisms that maintain L-PGDS and PGD2 levels. When L-PGDS secretion is blocked by botulinum toxin, extracellular PGD2 levels decrease, which induces an increase in *Ptgds* mRNA to promote PGD2 production and a decrease in *Hpgd* mRNA to reduce PGD2 inactivation.

Given the enriched expression of L-PGDS by oligodendrocyte-lineage cells (*Zhang et al., 2014*; *Zhang et al., 2016*) and its localization in the extracellular space (*Figure 7A and C*; *Hoffmann et al., 1993*), our results indicate that L-PGDS is an oligodendrocyte-lineage cell-secreted autocrine/paracrine molecule that promotes oligodendrocyte development and myelination. Our results are consistent with the following model: OPCs secrete autocrine/paracrine signals such as L-PGDS to promote oligodendrocyte development and myelination. When VAMP1/2/3-dependent exocytosis is blocked, L-PGDS secretion is defective, leading to defective oligodendrocyte development and myelination. Overexpression of *Ptgds* (L-PGDS) in exocytosis-deficient PD:ibot mice restores L-PGDS levels and partially rescues myelination defect. Our discovery of the role of exocytosis and L-PGDS in oligodendrocytes provides insight into the mechanisms regulating oligodendrocyte development and myelination and reveals novel molecular targets for future efforts aimed toward enhancing myelination for neural repair.

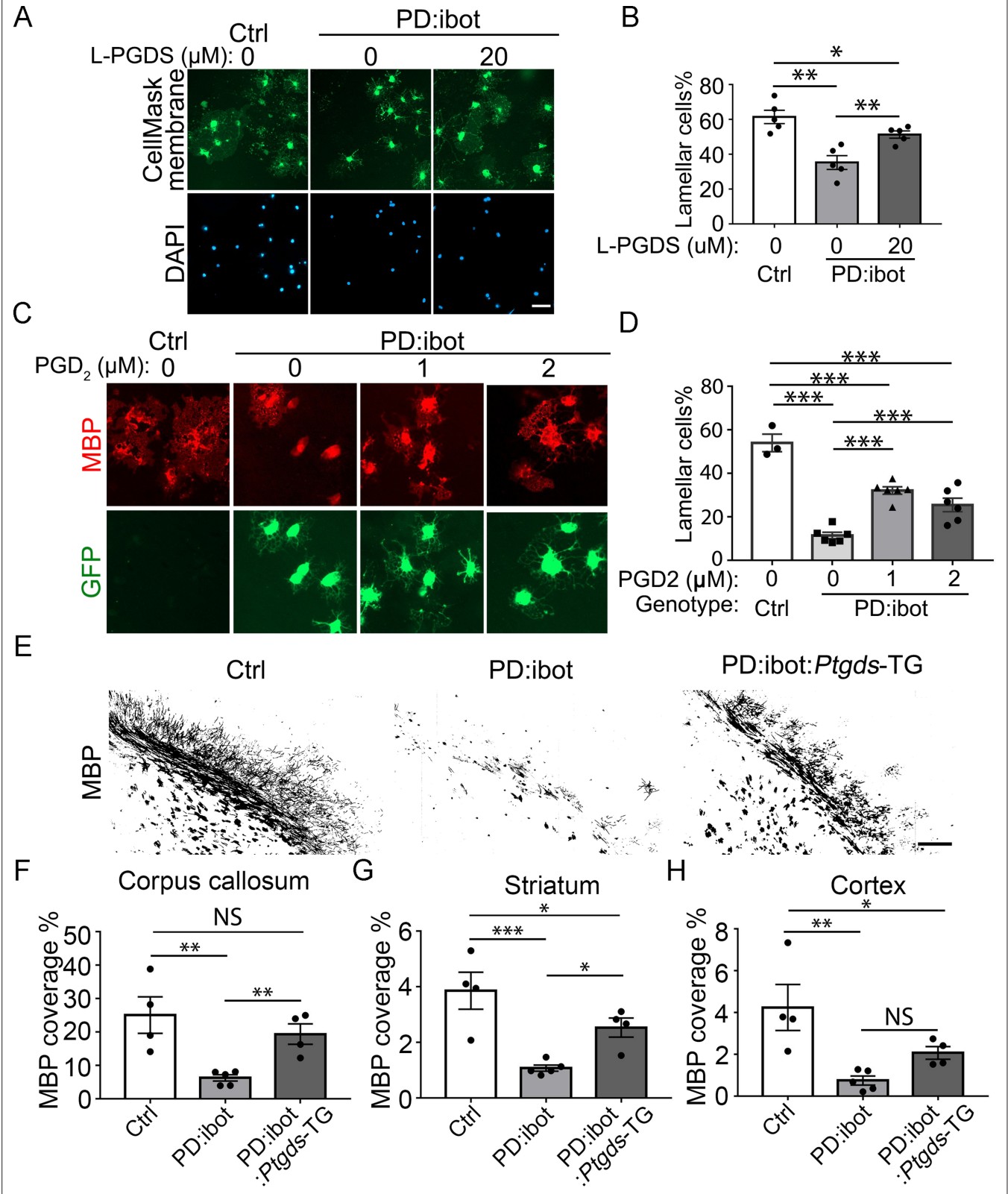

**Figure 9.** Rescue of oligodendrocyte deficit in PD:ibot by L-PGDS protein, PGD2 in vitro, and myelination deficit in PD:ibot by L-PGDS overexpressing transgenic mice. (**A**) Oligodendrocyte cultures from wild-type mice after 7 days of differentiation in the presence and absence of L-PGDS protein. Green: Membrane-staining version of CellMask, which labels the membranes of all cells. Blue: DAPI. Scale bars: 50 μm. (**B**) Quantification of cells with lamellar morphology. One-way ANOVA with Benjamini, Krieger, and Yekutieli's two-stage linear step-up FDR procedure for multiple comparisons.

*Figure 9 continued*

Every group was compared with every other group. N=5 cultures from 4 mice per group. Control vs. PD:ibot, L-PGDS 0 µM: p=0.0082; PD:ibot, L-PGDS 0 µM vs. PD:ibot, L-PGDS 20 µM: p=0.0082; control vs. PD:ibot, L-PGDS 20 µM: p=0.049. Lamellar cells%: Control: 61.5±3.9; PD:ibot, 0 µM L-PGDS: 35.3±4.0; PD:ibot, 20 µM L-PGDS: 51.3±2.1. (**C**) Partial rescue of oligodendrocyte differentiation by PGD2. Oligodendrocytes from PD:ibot and control mice in culture after 7 days of differentiation. Red: MBP immunofluorescence. Green: ibot-GFP, only present in cells from PD:ibot mice. Scale bars: 50 µm. (**D**) Quantification of the percentage of lamellar cells. One-way ANOVA with Benjamini, Krieger, and Yekutieli's two-stage linear step-up FDR procedure for multiple comparisons. Every group was compared with every other group. For control, N=3 cultures from 3 mice per group. For all others, N=6 cultures from 6 mice per group. wild type control vs. PD:ibot +DMSO: p=<0.0001; PD:ibot +DMSO vs. PD:ibot, PGD2 1 µM: p=<0.0001; PD:ibot +DMSO vs. PD:ibot, PGD2 2 µM: p=0.0001; wild type control vs. PD:ibot, PGD2 1 µM: p=<0.0001; wild type control vs. PD:ibot, PGD2 2 µM: p=<0.0001; PD:ibot, PGD2 1 µM vs. PD:ibot, PGD2 2 µM: p=0.01. Lamellar cells%: wild type control: 54.0±4.0, PD:ibot +DMSO; 11.4±1.5; PD:ibot +1 µM PGD2: 32.1±1.7, PD:ibot +2 µM PGD2: 25.5±3.1. (**E**) MBP immunofluorescence at P9 in control, PD:ibot, and PD:ibot:*Ptgds*-TG brains. Scale bar: 200 µm. (**F–H**) Quantification of MBP⁺ area in the the corpus callosum (**F**), striatum (**G**), and cortex (**H**) at P9. All control mice are ibot only. One-way ANOVA with Benjamini, Krieger, and Yekutieli's two-stage linear step-up FDR procedure for multiple comparisons. Every group was compared with every other group. For control and PD:ibot:*Ptgds*-TG, N=4 mice per group; For PD:ibot, N=5 mice per group. In corpus callosum, control vs. PD:ibot: p=0.0026; control vs. PD:ibot:*Ptgds*-TG, p=0.097; PD:ibot vs. PD:ibot:*Ptgds*-TG, p=0.0099. In striatum, control vs. PD:ibot: p=0.0006; control vs. PD:ibot:*Ptgds*-TG, p=0.017; PD:ibot vs. PD:ibot:*Ptgds*-TG, p=0.014. In cortex, control vs. PD:ibot: p=0.0049; control vs. PD:ibot:*Ptgds*-TG, p=0.040; PD:ibot vs. PD:ibot:*Ptgds*-TG, p=0.11. MBP coverage in the corpus callosum (%): 25.1±5.5 in control, 6.3±0.9 in PD:ibot, and 19.4±3.1 in PD:ibot:*Ptgds*-TG; MBP coverage in the striatum (%): 3.9±0.7 in control, 1.1±0.11 in PD:ibot, and 2.5±0.35 in PD:ibot:*Ptgds*-TG. MBP coverage in the cortex (%): 4.2±1.1 in control, 0.75±0.22 in PD:ibot, and 2.07±0.30 in PD:ibot:*Ptgds*-TG.

The online version of this article includes the following source data for figure 9:

**Source data 1.** It contains original data points for **Figure 9B, D, F, G and H**.

## Discussion

In this study, we showed that oligodendrocyte-lineage cell-secreted molecules promote oligodendrocyte development and myelination in an autocrine/paracrine manner. We identified L-PGDS as one such secreted molecule, thus revealing a novel cellular mechanism regulating oligodendrocyte development.

Previously, the roles of VAMP3 and related pathways in myelin protein delivery and oligodendrocyte morphogenesis have been investigated largely in vitro using cultures of an OPC-like cell line (Oli-Neu cells) or primary oligodendrocytes. For example, VAMP3 and VAMP7 knockdown inhibits the transport of a myelin protein, Proteolipid Protein1 in vitro (*Feldmann et al., 2011*). Tetanus toxin, which cleaves VAMP1/2/3, inhibits oligodendrocyte branching in vitro (*Sloane and Vartanian, 2007*). Syntaxin4, a potential binding partner of VAMP3, is required for the transcription of MBP in oligodendrocytes in vitro (*Bijlard et al., 2015*). Our study established a requirement of VAMP1/2/3-dependent exocytosis in oligodendrocyte development, myelination, and motor behavior in vivo and identified L-PGDS as an oligodendrocyte-lineage cell-secreted protein that promotes oligodendrocyte development and myelination.

VAMP1/2/3-dependent exocytosis is not the only pathway employed by oligodendrocyte-lineage cells to release molecules that mediate cell-cell interactions. For example, oligodendrocytes release exosome-like vesicles that inhibit the growth of myelin-like membranes in vitro (*Bakhti et al., 2011*). Tetanus toxin cleaves VAMP1/2/3 but does not affect exosome release (*Fader et al., 2009*). Therefore, the role of VAMP1/2/3-dependent exocytosis in promoting myelination and the effect of exosome-like vesicles in inhibiting myelination are likely parallel pathways independent of each other. In future studies, it could be interesting to determine the signals that regulate VAMP1/2/3-dependent exocytosis and VAMP1/2/3-independent exosome release during development and disease in vivo and thus define how these two seemingly opposing effects are coordinated to shape precise and dynamic myelination.

We observed a decrease in the percentage of PDGFRa⁺ OPCs expressing botulinum toxin-B from P8 to P30 in PD:ibot mice (***Figure 1G*** and ***Figure 1—figure supplement 2C***). Since OPC proliferation is increased and the overall OPC densities do not change in PD:ibot mice, a decrease in the percentage of OPCs expressing the toxin is consistent with the death of some toxin-expressing OPCs. Although our cleaved caspase-3 immunohistochemistry results did not show a difference in OPC survival between PD:ibot and control cells (***Figure 3—figure supplement 1***), we cannot exclude the possibilities that OPC die through non-apoptotic mechanisms or that microglia clear dead cells too rapidly for accurate counting.

OPCs are present throughout the CNS in adults (*Hughes et al., 2013*; *Kang et al., 2010*), even in demyelinated lesions in patients with multiple sclerosis (*Franklin, 2002*). Therefore, inducing oligodendrocyte and new myelin formation is an attractive strategy for treating demyelinating diseases. However, remyelination therapy has not been successful so far (*Franklin, 2002*), underscoring the need for a more complete understanding of the mechanisms regulating oligodendrocyte and myelin development. Our discovery of the role of L-PGDS in oligodendrocyte development and myelination adds to the knowledge of the molecular regulation of myelination. Interestingly, mixed results have been reported on the level of the *Ptgds* gene and the L-PGDS protein in multiple sclerosis patients and mouse models (*Jäkel et al., 2019*; *Kagitani-Shimono et al., 2006*; *Penkert et al., 2021*). During remyelination in mice, PGD2 levels increase (*Penkert et al., 2021*). Future studies should clarify the involvement of L-PGDS in multiple sclerosis and its therapeutic potential in promoting remyelination.

A recent study shows that the gene encoding L-PGDS, *Ptgds*, marks a subpopulation of OPCs more resilient to spinal cord injury than other OPCs (*Floriddia et al., 2020*). Thus, the function of L-PGDS in OPC heterogeneity and their responses to injury and other neurological disorders will be interesting to explore in the future.

The product of the L-PGDS enzyme, PGD2, binds and activates two G-protein-coupled receptors, DP1 and DP2 (Gpr44) (*Narumiya and Furuyashiki, 2011*). In addition, PGD2 undergoes non-enzymatic conversion to 15d-PGJ2, which activates the peroxisome proliferator-activated receptor-γ (*Scher and Pillinger, 2005*). Future studies should aim to identify the receptor(s) that mediates the effect of L-PGDS on oligodendrocyte development and myelination, as well as the downstream signaling pathways.

OPCs and oligodendrocytes express numerous genes encoding secreted proteins (*Zhang et al., 2014*). Although we identified the role of L-PGDS in oligodendrocyte development, our results do not rule out contributions from other secreted molecules. Our RNA-seq dataset provides a roadmap for future investigation of the roles of additional oligodendrocyte-lineage cell-secreted molecules in the brain.

Blocking exocytosis with botulinum toxin B may reduce the delivery of proteins and lipids to the plasma membrane, therefore causing cell-autonomous effects on oligodendrocyte development in addition to blocking secretion. Both cell-autonomous and cell-non-autonomous mechanisms may be involved in the effect of blocking exocytosis on oligodendrocyte development (*Fekete et al., 2022*; *Lam et al., 2022*). Our transwell rescue experiment and the comparisons between control cells in PD:ibot culture and control cells in control culture (*Figure 6*) support the importance of secreted molecules but do not rule out cell-autonomous mechanisms.

Two recent publications from the Zuchero and Nishiyama groups also utilized the ibot mice to investigate the role of VAMP2/3-mediated exocytosis in oligodendrocyte development and myelination (*Fekete et al., 2022*; *Lam et al., 2022*). Together with data presented here, the three complementary studies utilized different Cre lines to express botulinum toxin-B in oligodendrocyte lineage cells (*NG2-Cre* in Fekete et al., *CNP-Cre* in Lam et al., and *PDGFRa-CreER* in this study). Interestingly, all three ibot mouse strains exhibit myelination defects in the central nervous system (the spinal cord in Fekete et al., the brain in this study, and the brain and the spinal cord in Lam et al.). These consistent results from three mouse strains obtained by independent groups demonstrate an important role of VAMP2/3-mediated exocytosis in myelin development. VAMP2/3-mediated exocytosis may affect membrane expansion, secretion, and signaling. To provide mechanistic insight into the function of VAMP2/3 in myelin development, these three studies focused on different aspects of VAMP2/3 function. In *NG2*:ibot mice, Fekete et al. observed elevated levels of Fyn kinase, which is a signaling molecule implicated in regulating oligodendrocyte maturation and myelination. Using *CNP*:ibot mice, Lam et al. demonstrated the role of VAMP2/3-dependent exocytosis in membrane expansion and insertion of plasma membrane proteins during oligodendrocyte maturation, and further performed elegant imaging studies to capture exocytosis events of VAMP2/3-containing vesicles in oligodendrocytes in vitro and in vivo. In this study, we demonstrated the cell-non-autonomous function of VAMP2/3-dependent exocytosis in oligodendrocyte development and identified a novel role of a secreted molecule, L-PGDS, in oligodendrocyte development. Because VAMP2/3-mediated exocytosis may contribute to the membrane insertion and secretion of a variety of molecules, which may in turn activate multiple intracellular signaling pathways, it is not surprising that multiple mechanisms mediate the effect of VAMP2/3 in oligodendrocyte development and myelination. These three

complementary studies provide key insight toward a comprehensive understanding of the role of VAMP2/3 in myelination.

Although all three ibot mouse strains exhibit myelin defects, other phenotypes differ among the strains. For example, the ibot mice in Fekete et al. and this study have reduced oligodendrocyte densities, whereas Lam et al. did not observe an overt loss of oligodendrocytes. We used the CC1 antibody to detect differentiated oligodendrocytes and Fekete et al. used CC1 (i.e. QKI7) and several additional markers to validate a reduction in mature oligodendrocytes (ASPA, GST-p, Nkx2.2). Differences in the timing of botulinum toxin-B expression driven by the Cre lines may account for the divergent results. Fekete et al. used *NG2-Cre* and this study used *PDGFRa-CreER*, both Cre lines drive toxin expression in OPCs, whereas Lam et al. used *CNP-Cre* that predominantly drives toxin expression in oligodendrocytes. The intriguing divergent phenotypes suggest the possibility of a stage-dependent requirement of VAMP2/3-mediated exocytosis in the development of oligodendrocyte-lineage cells, which may be investigated in the future to clarify the complex roles of VAMP2/3 in oligodendrocyte development and myelination. Together, the three complementary studies revealed the importance of VAMP2/3-mediated exocytosis in myelination through a combination of cell-autonomous and cell-non-autonomous mechanisms.

## Materials and methods
### Lead contact and materials availability
Further information and requests for resources and reagents should be directed to and will be fulfilled by the Lead Contact, Ye Zhang (yezhang@ucla.edu). This study did not generate new unique reagents.

### Experimental animals
All animal experimental procedures (protocols: #R-16–079 and #R-16–080) were approved by the Chancellor's Animal Research Committee at the University of California, Los Angeles, and conducted in compliance with national and state laws and policies. All the mice were group-housed in standard cages (maximum 5 mice per cage). Rooms were maintained on a 12 hr light/dark cycle. *Pdgfra-CreER* (Jax #018280), ibot (Jax #018056), *Gfap*-Cre (Jax #024098), and *Tek*-Cre (Jax, #004128) mouse strains were obtained from Jackson Laboratories. L-PGDS global knockout (*Ptgds*$^{-/-}$) mouse strain was originally from Urade, cryopreserved by Garret FitzGerald (*Urade and Hayaishi, 2000*). L-PGDS overexpressing transgenic mouse strain (*Ptgds*-TG, line B7) was originally from Urade, cryopreserved by JCRB Laboratory Animal Resource Bank (*Pinzar et al., 2000*). All PD:ibot and control mice, including wildtype, Cre-only and ibot-only mice used were congenic. We used PD:ibot and littermate control mice raised under the same maternal care in the same cage to minimize environmental confounding factors.

### OPC purification and culture
Whole brains excluding the olfactory bulbs and the cerebellum from one pup at postnatal day 7 to day 8 were used to make each batch of OPC culture. OPCs were purified using an immunopanning method described before (*Emery and Dugas, 2013*). Briefly, the brains were digested into single-cell suspensions using papain. Microglia and differentiated oligodendrocytes were depleted using anti-CD45 antibody- (BD Pharmingen, cat #550539) and GalC hybridoma-coated panning plates, respectively. OPCs were then collected using an O4 hybridoma-coated panning plate. For most culture experiments, cells were plated on 24-well plates at a density of 30,000 per well. For comparison of OPC differentiation at different densities, OPCs were plated at densities of 5000 per well and 40,000 per well. For all experiments, OPCs were first kept in proliferation medium containing growth factors PDGF (10 ng/ml, Peprotech, cat #100–13 A), CNTF (10 ng/ml, Peprotech, cat #450–13), and NT-3 (1 ng/ml, Peprotech, cat #450–03) for 2–3 days, and then switched to differentiation medium containing thyroid hormone (40 ng/ml, Sigma, cat #T6397-100MG) but without PDGF or NT-3 for seven days to differentiate them into oligodendrocytes as previously described (*Emery and Dugas, 2013*). Half of the culture media was replaced with fresh media every other day. All the cells were maintained in a humidified 37°C incubator with 10% $CO_2$. Cells from both female and male mice were used. For coculture experiments with inserts, OPCs were purified from PD:ibot and littermate control mice as described above. A total of 100,000 cells per well were plated on inserts with 1 μm

diameter pores (VWR, cat #62406–173), and the inserts were placed on top of wells with cells plated at 30,000 cells per well density on 24-well culture plates. 200 µl medium was added per insert and 500 µl medium was added per well under the inserts.

## Drugs and treatment

4-Hydroxy-tamoxifen stock solutions were made by dissolving 4-hydroxy-tamoxifen (Sigma, H7904) into pure ethanol at 10 mg/ml. The stock solutions were stored at –80°C until use. On the day of injection, an aliquot of 4-hydroxy-tamoxifen stock solution (100 µl) was thawed and mixed with 500 µl sunflower oil by vortexing for 5 min. Ethanol in the solution was vacuum evaporated in a desiccator (VWR, 89054–050) for an hour. 0.1 mg 4-hydroxy-tamoxifen was injected into each mouse subcutaneously daily for 2 days at P2 and P4. An L-PGDS inhibitor, AT-56 (Cayman Chemicals, cat #13160), and prostaglandin D2 (Cayman Chemicals, cat #12010) were dissolved in dimethyl sulfoxide (DMSO). To inhibit L-PGDS activity in vitro, AT-56 was added to the oligodendrocyte culture medium at 1 µM and 5 µM every other day. HPGD protein (R&D system, cat#5660-DH-010) was added to the oligodendrocyte culture medium at 3 µM and 6 µM daily. L-PGDS protein (Cayman Chemicals, cat#10006788) was added to the oligodendrocyte culture at 20 µM daily. For prostaglandin D2 treatment, prostaglandin D2 was added to the oligodendrocyte culture medium at 1 µM and 2 µM every 12 hr. An equal amount of DMSO was added to the control wells. Because a metabolite of prostaglandin D2, 15-d-prostaglandin J2, induces cell death, which can be prevented by N-acetyl cysteine (*Lee et al., 2008*), we included 1 mM N-acetyl cysteine, which is shown to improve cell survival, in the culture media of prostaglandin D2-treated and control cells.

## Purification of microglia, oligodendrocytes, oligodendrocyte precursor cells, and astrocytes

Whole brains excluding the olfactory bulbs and the cerebellum from P17 PD:ibot and control littermates were used for the purification of microglia, oligodendrocytes, oligodendrocyte precursor cells, and astrocytes. A single-cell suspension was prepared as described above. Cells were incubated for 30 min on an anti-CD45 antibody (BD Pharmingen, cat#550539, 1.25 µg/ml)-coated panning plate to harvest microglia, followed by two sequential CD45-coated panning plates to deplete remaining microglia. Cell suspension was then incubated for 30 min on a GalC hybridoma-coated panning plate to collect differentiated oligodendrocytes, followed by two more GalC hybridoma-coated plates to deplete any remaining differentiated oligodendrocytes. The remaining cells were incubated for 30 min on an O4 hybridoma-coated panning plate to collect oligodendrocyte precursor cells, followed by two O4 hybridoma-coated panning plates to deplete remaining OPCs. The cell suspension was then incubated with an anti-HepaCAM antibody (R&D Systems, cat# MAB4108)-coated panning plate to collect astrocytes.

## RNA-seq

Total RNA was extracted using the miRNeasy Mini kit (Qiagen cat #217004). The concentrations and integrities of the RNA were measured using TapeStation (Agilent) and Qubit. Sixty ng total RNA from each sample was used for library preparation. cDNA was generated using the Nugen Ovation V2 kit (Nugen) and fragmented using the Covaris sonicator. Sequencing libraries were prepared using the Next Ultra RNA Library Prep kit (New England Biolabs) with 12 cycles of PCR amplification. An Illumina HiSeq 4000 sequencer was used to sequence these libraries and each sample had an average of 19.1 ± 2.9 million 50 bp single-end reads.

## RNA-seq data analysis

STAR package was used to map reads to mouse genome mm10 and HTSEQ was used to obtain raw counts from sequencing reads. EdgeR-Limma-Voom packages in R were used to calculate Reads per Kilobase per Million Mapped Reads (RPKM) values from raw counts. DESeq2 package was used to analyze differential gene expression.

## Immunohistochemistry and immunocytochemistry

Mice were anesthetized with isoflurane and transcardially perfused with phosphate-buffered saline (PBS) followed by 4% paraformaldehyde (PFA). Brains were removed and post-fixed in 4% PFA at

4 °C overnight. Brains were washed with PBS and cryoprotected in 30% sucrose at 4 °C for 2 days before embedding in optimal cutting temperature compound (Fisher, cat #23-730-571) and stored at –80 °C. Brains were sectioned using a cryostat (Leica) into 30-μm-thick sections and floating sections were blocked and permeabilized in 5% donkey serum with 0.3% Tween-20 in PBS and then stained with primary antibodies against GFP (Aves Labs, Inc, cat #GFP-1020, dilution 1:500), PDGFRα (R&D Systems, cat #AF1062, dilution 1:500), Olig2 (Millipore, cat #211F1.1, dilution 1:500), CC1 (Millipore, cat #OP80, dilution 1:500), MBP (Abcam, cat #ab7349, dilution 1:500), Iba1 (FUJIFILM Wako, cat# 019–19741, dilution 1:1000), SMI-32 (BioLegend, cat# 801702, dilution 1:500), and cleaved caspase-3 (Cell Signaling, cat #9661 S, dilution 1:500) at 4 °C overnight. Heat-induced epitope retrieval is performed for sections stained with SMI-32. All other staining did not require antigen retrieval. Sections were washed three times with PBS and incubated with fluorescent secondary antibodies (Invitrogen) at 4 °C overnight. Sections were mounted onto Superfrost Plus micro slides (Fisher, cat #12-550-15) and covered with mounting medium (Fisher, cat #H1400NB) and glass coverslips. Slides were imaged with a Zeiss Apotome epifluorescence microscope. Adjacent regions are tiled using Zeiss tiling function. Tiling borders in some representative images are caused by differing background intensities where different tiles are stitched together and are removed using 'Substract Background' function in Fiji with the setting (rolling ball radius = 50 pixels).

A large area of cerebral cortex, including motor cortex and somatosensory cortex, from bregma 1 mm to –2 mm was used for the following quantification: MBP coverage in the cortex, the density of oligodendrocyte-lineage marker genes (CC1, Olig2, PDGFRα, *Enpp6*, and *Mbp*), and the density of apoptotic cells (cleaved caspase-3[+]).

For immunocytochemistry of cultured cells, cells were fixed with 4% PFA and 0.3% Tween-20 in PBS. After blocking in 5% donkey serum, cells were then stained with the primary antibodies described above, VAMP2 (Synaptic Systems, cat #104 211, dilution 1:100), and VAMP3 (Synaptic Systems, cat #104 103, dilution 1:500) at 4 °C overnight. After three washes in PBS, cells were stained with secondary antibodies and the cytoplasm-staining CellMask Blue (Invitrogen, cat #H32720, 1:1,000) at 4 °C overnight. To stain the cells with the membrane-staining CellMask (Invitrogen, cat#C37608 Dilution 1:1000), we added the dye into the culture medium and incubate it with the cells for 13 min at 37 °C. After staining, cells were washed once with PBS and fixed with 4% PFA for 15 min at room temperature. Cells were washed three times with PBS before covered with mounting medium (Fisher, cat #H1400NB). Slides were imaged with a Zeiss Apotome epifluorescence microscope.

Fluorescence microscopy images were cropped, and brightness contrast was adjusted with identical settings across genotype, treatment, and control groups using Photoshop and ImageJ. All the images were randomly renamed using the following website (https://www.random.org/) and quantified with the experimenter blinded to the genotype and treatment condition of the samples. Cells with MBP[+] membrane spreading out were identified as lamellar cells. Illustrations were made with Biorender.

## RNAscope in situ hybridization

P15 mice were transcardially perfused with PBS followed by 4% PFA. Brains were removed and post-fixed in 4% PFA for an additional 2 hr at room temperature and then overnight at 4 °C. Tissues were then dehydrated in 30% sucrose, embedded in OCT compound, and cut into 20- to 30-μm-thick sections. RNAscope Multiplex Fluorescent Reagent Kit v2 (ACDBio, cat# 323100) was used per manufacturer's protocol. Probes used in this paper were purchased from ACDBio for mouse *Enpp6* (cat#511021-C2) and *Mbp* (cat# 451491). Slides were imaged with a Zeiss Apotome epifluorescence microscope at equal power and exposure across all samples stained with the same set of probes.

## Transmission electron microscopy

Brain specimens for transmission electron microscopy were prepared as described before (*Salazar et al., 2018*). Mice were anesthetized using isoflurane and transcardially perfused with 0.1 M phosphate buffer (PB) followed by 4% PFA with 2.5% glutaraldehyde in 0.1 M PB buffer. Brains were removed and post-fixed in 4% PFA with 2.5% glutaraldehyde in 0.1 M PB for another two days. Brains were sliced with Young Mouse Brain Slicer Matrix (Zivic Instruments, cat #BSMYS001-1) and a small piece of the corpus callosum was isolated from brain sections at 0–1 mm anterior to Bregma. After washing, samples were then post-fixed in 1% osmium tetroxide in 0.1 M PB (pH 7.4) and dehydrated

through a graded series of ethanol concentrations. After infiltration with Eponate 12 resin, samples were embedded in fresh Eponate 12 resin and polymerized at 60 °C for 48 hr. Ultrathin sections of 70 nm thickness were prepared and placed on formvar/carbon-coated copper grids and stained with uranyl acetate and lead citrate. Grids were examined using a JEOL 100 CX transmission electron microscope at 60 kV and images were captured by an AMT digital camera (Advanced Microscopy Techniques Corporation, model XR611) by the Electron Microscopy Core Facility, UCLA Brain Research Institute. Before analysis, TEM images were blinded as described above. An axon that is encircled by 2 or more compacted layers of electron-dense lines is defined as a myelinated axon. *g-ratio* was determined by dividing the mean diameter of the area inside myelin by the mean diameter of the same axon with myelin. Approximately 100 myelinated axons from each group were analyzed for *g-ratio* analysis. For measuring the percentage of myelinated axons, approximately 900 axons from three mice were used for each group. More than 1300 axons from four mice from each group were used to measure axon density and approximately 500 axons from four mice from each group were used for axon diameter measurement.

## RNA extraction and qPCR

100 k Day 7 differentiated oligodendrocytes were lysed with the Qiazol lysis buffer and RNA was extracted using the PureLink RNA mini kit (Invitrogen 12183018 A) following the manufacturer's protocol. cDNA was generated using the SuperScript III First-Strand Synthesis SuperMix (Invitrogen 18080400). The primer for *Mbp* was designed using the primerblast tool from NCBI. Primerblast was used to validate the specificity of primers. The sequences for *Mbp* and *Gapdh* primers were provided in the key resource table. PowerUp SYBR Green Master Mix (Applied Biosystems A25742) and a QuantStudio 3 Real-Time PCR System (Thermo fisher, cat# A28567) were used for qRT-PCR reaction.

## Western blot

We purified OPCs from PD:ibot and control mice by immunopanning and cultured them in proliferation medium for 2–3 days and differentiation medium for 7 days as described above. To collect secreted samples, culture media were mixed with ethylenediaminetetraacetic acid (EDTA)-free protease inhibitor cocktail (Sigma, cat #4693159001) at a 6:1 ratio and centrifuged at 1000×*g* for 10 min to remove dead cells and debris. To collect whole-cell lysates, cells were washed with PBS, lysed with radioimmunoprecipitation assay buffer containing EDTA-free protease inhibitor cocktail, and centrifuged at 12,000×*g* for 10 min to remove cell debris.

All samples were mixed with sodium dodecyl sulfate (SDS) sample buffer (Fisher, cat # AAJ60660AC) and 2-mercaptoethanol before boiling for 5 min. Samples were separated by SDS-polyacrylamide gel electrophoresis, followed by transferring to polyvinylidene difluoride membranes via wet transfer at 300 mA for 1.5 hr. Membranes were blocked with clear milk blocking buffer (Fisher, cat #PI37587) for 1 hr at room temperature and incubated with primary antibodies against L-PGDS (Santa Cruz Biotechnology, cat #sc-390717, dilution 1:1000), GAPDH (Sigma, cat #CB1001, dilution 1:5000), BoNT-B Light Chain (R&D Systems, cat #AF5420-SP, dilution 1:1000), VAMP2 (Synaptic Systems, cat #104 211, dilution 1:1000), VAMP3 (Novus Biological, cat # NB300-510-0.025mg, dilution 1:1000) and MBP (Abcam, cat #ab7349, dilution 1:1000) at 4 °C overnight. Membranes were washed with tris-buffered saline with Tween 20 (TBST) three times and incubated with horseradish peroxidase-conjugated secondary antibodies (Mouse, Cell Signaling, cat #7076 S; Rabbit, Cell Signaling, cat #7074 S; Rat, Cell Signaling, cat #7077 S; Sheep, Thermo Fisher, cat #A16041) or Donkey anti-Mouse IgG (H+L) Highly Cross-Adsorbed Secondary Antibody, Alexa Fluor Plus 647 (Fisher, cat # PIA32787,1:1000) (only for GAPDH in *Figure 5B*) for 1 hr at room temperature. After three washes in TBST buffer, SuperSignal West Femto Maximum Sensitivity Substrate (Fisher, cat #PI34095) was added to the membranes, and the signal was visualized using a ChemiDoc MP Imaging system (BIO-RAD).

## Motor behavior

Mice were familiarized with being picked up and handled by the experimenter daily for three days before the test to reduce stress. Mice were also habituated to the rotarod testing room for 15 min prior to all testing. Both male and female adult mice (2–5 months old) were used in the rotarod test. Mice were given three trials per day for three consecutive days (5–60 rpm over 5 min, with approximately

30 min between successive trials). The latency to fall was measured and the experimenter was blinded to the genotype of the mice during the test.

## Quantification and statistical analysis

The numbers of animals and replicates are described in the figures and figure legends. The RNA-seq data were analyzed using the DESeq2 package. Adjusted p-values smaller than 0.05 were considered significant. For all non-RNA-seq data, analyses were conducted using Prism 8 software (Graphpad). The normality of data was tested by the Shapiro-Wilke test. For data with a normal distribution, Welch's t-test was used for two-group comparisons and one-way ANOVA was used for multi-group comparisons. An estimate of variation in each group is indicated by the standard error of the mean (S.E.M.). * $p<0.05$, ** $p<0.01$, *** $<0.001$. An appropriate sample size was determined when the study was being designed based on published studies with similar approaches and focus as our study. A biological replicate is defined as one mouse. Different culture wells from the same mouse or different images taken from the brains or cell cultures from the same mouse are defined as technical replicates. All statistical tests were performed with each biological replicate/mouse as an independent observation. The number of times each experiment was performed is indicated in figure legends. No data were excluded from the analyses. Mice and cell cultures were randomly assigned to treatment and control groups. Imaging analyses and behavior tests were conducted when the experimenter was blinded to the genotypes or treatment conditions.

## Code availability

This study did not generate new codes.

## Acknowledgements

We thank Akiko Nishiyama, J Bradley Zuchero, Hui Zong, and Mable Lam for their advice and editing of our manuscript. We thank Qingyun Li, Richard Breyer, Henry Lin, and Ginger Milne for their advice. We thank Garret FitzGerald for reagents. We thank the Eli and Edythe Broad Center of Regenerative Medicine and Stem Cell Research, UCLA BioSequencing Core Facility for their services, Mahnaz Akhavan and Suhua Feng for their technical support. This work is supported by the Knaub Postdoctoral Fellowship to LP, the National Institute of Neurological Disorders and Stroke of the National Institute of Health (NIH) R00NS089780, R01NS109025, R01NS099102 (CT), the National Institute of Aging of the NIH R03AG065772, the National Institute of Child Health and Human Development P50HD103557, the National Center for Advancing Translational Science UCLA CTSI Grant UL1TR001881, the W M Keck Foundation Junior Faculty Award, the UCLA Eli and Edythe Broad Center of Regenerative Medicine and Stem Cell Research Innovation Award, the Ablon Faculty Scholar Award, and the Friends of the Semel Institute for Neuroscience & Human Behavior Friends Scholar Award to YZ.

## Additional information

### Competing interests

Ye Zhang: consulted for Ono Pharmaceutical. The other authors declare that no competing interests exist.

### Funding

| Funder | Grant reference number | Author |
| --- | --- | --- |
| UCLA Brain Research Institute | Knaub Postdoctoral Fellowship | Lin Pan |
| National Institute of Neurological Disorders and Stroke | R00NS089780 | Ye Zhang |
| National Institute on Aging | R03AG065772 | Ye Zhang |

| Funder | Grant reference number | Author |
|---|---|---|
| National Institute of Child Health and Human Development | P50HD103557 | Ye Zhang |
| National Center for Advancing Translational Science UCLA CTSI Grant | UL1TR001881 | Ye Zhang |
| W. M. Keck Foundation | W. M. Keck Foundation junior faculty award | Ye Zhang |
| UCLA Eli and Edythe Broad Center of Regenerative Medicine and Stem Cell Research | Innovation Award | Ye Zhang |
| Wendy Ablon Foundation | Ablon Scholar Award | Ye Zhang |
| National Institute of Neurological Disorders and Stroke | R01NS109025 | Ye Zhang |
| Friends of the Semel Institute for Neuroscience & Human Behavior | Friends scholar award | Ye Zhang |

The funders had no role in study design, data collection and interpretation, or the decision to submit the work for publication.

## Author contributions

Lin Pan, Ye Zhang, Conceptualization, Resources, Data curation, Formal analysis, Supervision, Funding acquisition, Validation, Investigation, Visualization, Methodology, Writing - original draft, Project administration, Writing – review and editing; Amelia Trimarco, Resources, Investigation, Methodology, Provided and helped to analyze the brain samples of Ptgds knockout mice; Alice J Zhang, Formal analysis, Investigation, Assisted in mouse colony management and immunohistochemistry experiments; Ko Fujimori, Resources, Provided the Ptgds knockout mice and transgenic mice; Yoshihiro Urade, Resources, Provided the Ptgds knockout mice and transgenic mice; Lu O Sun, Conceptualization, Methodology, Writing – review and editing, Contributed to experimental design; Carla Taveggia, Conceptualization, Resources, Supervision, Methodology, Writing – review and editing

## Author ORCIDs

Lin Pan ⓘ http://orcid.org/0000-0001-8510-0032
Carla Taveggia ⓘ http://orcid.org/0000-0002-6531-9544
Ye Zhang ⓘ http://orcid.org/0000-0002-1546-5930

## Ethics

All animal experimental procedures (protocols: #R-16-079 and #R-16-080) were approved by the Chancellor's Animal Research Committee at the University of California, Los Angeles, and conducted in compliance with national and state laws and policies.

## Decision letter and Author response

Author response https://doi.org/10.7554/eLife.77441.sa2

## Additional files

### Supplementary files

• Supplementary file 1. Gene expression (RPKM) of OPCs, oligodendrocytes, microglia, and astrocytes from PD:ibot and littermate control mice at P17 determined by RNA-seq. Reads per kilobase of transcripts per million mapped reads (RPKM) are shown.

• Supplementary file 2. Differentially expressed genes in OPCs, oligodendrocytes, microglia, and astrocytes from PD:ibot and littermate control mice at P17. Genes with adjusted $P$-values <0.05 are shown. We used DESeq2 to determine differential gene expression.

• Supplementary file 3. Gene ontology terms associated with differentially expressed genes in

OPCs, oligodendrocytes, microglia, and astrocytes from PD:ibot and littermate control mice at P17. No gene ontology terms were significantly enriched in genes downregulated in OPCs, astrocytes, or upregulated in microglia in PD:ibot mice.

• Transparent reporting form

### Data availability

We deposited all RNA-seq data to the Gene Expression Omnibus under accession number GSE168569.

The following dataset was generated:

| Author(s) | Year | Dataset title | Dataset URL | Database and Identifier |
|---|---|---|---|---|
| Pan L, Trimarco A, Zhang AJ, Fujimori K, Urade Y, Sun LO, Taveggia C, Zhang Y | 2022 | Oligodendrocyte-lineage cell exocytosis and L-type prostaglandin D synthase promote oligodendrocyte development and myelination | https://www.ncbi. nlm.nih.gov/geo/ query/acc.cgi?acc= GSE168569 | NCBI Gene Expression Omnibus, GSE168569 |

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

# Appendix 1

## Appendix 1—key resources table

| Reagent type (species) or resource | Designation | Source or reference | Identifiers | Additional information |
|---|---|---|---|---|
| Strain, strain background (C57BL/6 x SJL) | *Pdgfra-CreER* | Jackson Laboratories | Cat#: 018280 | |
| Strain, strain background (FVB/N) | ibot | Jackson Laboratories | Cat# 018056 | |
| Strain, strain background (BALB/c x C57BL/6 NHsd) | *Gfap*-Cre | Jackson Laboratories | #024098 | |
| Strain, strain background (C3H x C57BL/6) | *Tek*-Cre | Jackson Laboratories | #004128 | |
| Strain, strain background (C57BL/6) | *Ptgds*$^{-/-}$ | **Urade and Hayaishi, 2000** | | originally from Urade, cryopreserved by Garret FitzGerald |
| Strain, strain background (FVB/N) | Human *Ptgds*-TG mice | **Pinzar et al., 2000** | | originally from Urade, cryopreserved by JCRB Laboratory Animal Resource Bank |
| Antibody | Anti-GFP antibody, host (chicken), Polyclonal | Aves Labs | Cat #GFP-1020; RRID: AB_10000240 | IHC: 1:500 ICC: 1:1000 |
| Antibody | Anti-PDGFRα antibody, host (goat), Polyclonal IgG | R&D Systems | Cat# AF1062; RRID: AB_2236897 | IHC: 1:500 |
| Antibody | Anti-Olig2 antibody, host (mouse), monoclonal IgG | Millipore | Cat# MABN50; RRID: AB_10807410 | IHC: 1:500 |
| Antibody | Anti-APC (Ab-7) (CC-1) antibody, host (mouse), monoclonal IgG | Millipore | Cat# OP80; RRID: AB_2057371 | IHC: 1:500 |
| Antibody | Anti-Myelin Basic Protein Antibody, host (rat), monoclonal | Abcam | Cat# ab7349; RRID: AB_305869 | IHC: 1:500 ICC: 1:1000 WB: 1:1000 |
| Antibody | Anti-cleaved caspase-3 (Asp175) Antibody, host (rabbit), polyclonal | Cell Signaling Technology | Cat# 9661; RRID: AB_2341188 | IHC: 1:500 ICC: 1:1000 |
| Antibody | Anti-Ki-67 Antibody (SP6), Antibody, host (rabbit), monoclonal | Thermo Fisher Scientific | Cat# MA5-14520; RRID: AB_10979488 | IHC: 1:500 |
| Antibody | Anti-C. botulinum BoNT-B Light Chain Antibody, host (sheep), polyclonal | R&D Systems | Cat# AF5420; RRID: AB_2044644 | IHC: 1:500 WB: 1:1000 |
| Antibody | Anti-PGD2 synthase Antibody (F-7), host (mouse), monoclonal IgG | Santa Cruz Biotechnology | Cat# sc-390717; RRID: AB_2800545 | IHC: 1:500 WB: 1:1000 |
| Antibody | Anti- GADPH antibody, host (mouse), monoclonal IgG | Millipore | Cat# CB1001; RRID: AB_2107426 | WB: 1:1000 |
| Antibody | Anti-CD45 antibody, host (rat), monoclonal IgG | BD Pharmingen | Cat#550539; RRID: AB_2174426 | Panning: 1.25 µg/ml |
| Antibody | Anti-HepaCAM antibody, host (mouse), monoclonal IgG | R&D Systems | Cat# MAB4108; RRID: AB_2117687 | Panning: 1 µg/ml |
| Antibody | Anti-VAMP3 antibody, host (rabbit), polyclonal | Novus Biological | Cat # NB300-510-0.025mg | WB: 1:1000 |
| Antibody | Anti-VAMP2 antibody, host (mouse), monoclonal IgG | Synaptic Systems | Cat #104 211; RRID: AB_887811 | ICC: 1:100 WB: 1:1000 |
| Antibody | Anti-VAMP3 antibody, host (rabbit), polyclonal | Synaptic Systems | Cat #104 103; RRID: AB_887812 | ICC: 1:500 |
| Antibody | Anti-Iba1 antibody, host (rabbit) | FUJIFILM Wako | Cat# 019–19741; RRID: AB_839504 | IHC: 1:1000 |
| Antibody | Anti-SMI-32 antibody, host (mouse), monoclonal | BioLegend | Cat# 801702; RRID: AB_2715852 | IHC: 1:500 |

*Appendix 1 Continued on next page*

*Appendix 1 Continued*

| Reagent type (species) or resource | Designation | Source or reference | Identifiers | Additional information |
|---|---|---|---|---|
| Chemical compound, drug | AT-56 | Cayman Chemicals | Cat#13160; CAS: 162640-98-4 | See Materials and Methods, Section "Drug and treatment". |
| Chemical compound, drug | Prostaglandin D2 | Cayman Chemicals | Cat# 12010; CAS: 41598-07-6 | See Materials and Methods, Section "Drug and treatment". |
| Chemical compound, drug | Thyroid hormone | Sigma | Cat #T6397-100MG | 40 ng/ml for induction of OPC differentiation |
| Chemical compound, drug | 4-hydroxy-tamoxifen | Sigma | H7904-25mg | See Materials and Methods, Section "Drug and treatment". |
| Peptide, recombinant protein | PDGF | Peprotech | Cat #100–13 A | See Materials and Methods, Section "OPC purification and culture". |
| Peptide, recombinant protein | CNTF | Peprotech | Cat #450–13 | See Materials and Methods, Section "OPC purification and culture". |
| Peptide, recombinant protein | NT-3 | Peprotech | Cat #450–03 | See Materials and Methods, Section "OPC purification and culture". |
| Peptide, recombinant protein | HPGD protein | R&D system | Cat#5660-DH-010 | See Materials and Methods, Section "Drug and treatment". |
| Peptide, recombinant protein | L-PGDS protein | Cayman Chemicals | Cat#10006788 | See Materials and Methods, Section "Drug and treatment". |
| Software, algorithm | ImageJ | *Schneider et al., 2012* | | https://imagej.nih.gov/ij/ |
| Software, algorithm | Prism 8 | Graphpad | | https://www.graphpad.com/scientific-software/prism/ |
| Software, algorithm | Photoshop | Adobe | | https://www.adobe.com/products/photoshop/free-trial-download.html |
| Software, algorithm | STAR package | *Dobin et al., 2013* | | https://github.com/alexdobin/STAR |
| Software, algorithm | HTSEQ | *Anders et al., 2015* | | https://htseq.readthedocs.io/en/release_0.11.1/count.html |
| Software, algorithm | EdgeR-Limma-Voom | *Law et al., 2016* | | http://bioconductor.org/packages/release/workflows/html/RNAseq123.html |
| Software, algorithm | DESeq2 | *Love et al., 2014* | | http://bioconductor.org/packages/release/bioc/vignettes/DESeq2/inst/doc/DESeq2.html |
| Sequence-based reagent | Mbp_F | This paper | PCR primers | 5'-TCACAGCGATCCAAGTACCTG-3' |
| Sequence-based reagent | Mbp_R | This paper | PCR primers | 5'-CCCCTGTCACCGCTAAAGAA-3' |
| Sequence-based reagent | Gapdh_F | This paper | PCR primers | 5'-TATGTCGTGGAGTCTACTGGTGTCTTC-3' |
| Sequence-based reagent | Gapdh_R | This paper | PCR primers | 5'-GTTGTCATATTTCTCGTGGTTCACACCC-3' |
| Other | CellMask stain | Invitrogen | Cat# H32720 | ICC: 1:1000 |
| Other | CellMask membrane stain | Invitrogen | Cat#C37608 | ICC: 1:1000 |
| Other | GalC hybridoma | From Ben A. Barres's lab | | Pannning: 1:4 dilution |
| Other | O4 Hybridoma | From Ben A. Barres's lab | | Pannning: 1:4 dilution |

