## [Editor Report]

The manuscript will be of interest to glial and myelin disease researchers. The well-designed combination of in vitro and in vivo approaches uncovers a potential mechanism of autocrine/paracrine signaling in oligodendrocyte maturation which provides an exciting avenue for future investigation.

---

## [Author Response]

Essential revisions:(1) It is unclear from the data presented whether PD:iBot cells are unable to differentiate and are therefore more likely to die. Experiments to test whether or not botulinum-expressing cells contribute to the population of surviving, differentiated oligodendrocytes are needed.

We thank the reviewers for raising a key point in characterizing the consequence of botulinum toxin expression in oligodendrocyte-lineage cells. We analyzed the overlap between GFP^+^ botulinum-expressing cells and the population of differentiated oligodendrocytes (Olig2^+^PDGFRa^-^CC1^+^ cells) and found that botulinum-expressing cells can survive and become differentiated oligodendrocytes (Figure 3—figure supplement 2, text page 12). Additionally, we performed a more thorough analysis of activated caspase-3^+^ apoptotic cells than was included in first submission and did not detect statistically significant differences between PD:ibot and control mice (Figure 3—figure supplement 1, text page 12).

(2) The manuscript's key experiment is the transwell co-culture study. First, raw data should be shown for this experiment.

We included raw data for this experiment (Figure 6G, text page 23)

Second, appropriate statistical analyses should be performed.

As Reviewer#2 suggested, we performed one-way ANOVA comparing every group to every other group with multiple comparison tests (Figure 6E-H text page 23).

Third, a non-MBP-reliant membrane marker should be employed for analysis.

We thank the reviewers for the insightful suggestion. We used the membrane version of CellMask suggested by Reviewer#3 and repeated the transwell co-culture experiment. The results are consistent with the results based on MBP (Figure 6—figure supplement 1, text page 24). In addition, we used the membrane version of CellMask for all the new cell culture experiments described below (L-PGDS rescue, HPGD etc.).

To further assess whether cell non-autonomous mechanisms contribute to the oligodendrocyte development defect in PD:ibot mice, we performed additional analysis in culture. We took advantage of the fact that all OPCs purified from PD:ibot mice are not botulinum-GFP-expressing (efficiency ~65% Figure 6B, text page 21). The GFP^-^ cells in PD:ibot OPC cultures do not express botulinum toxin and are competent in exocytosis. We compared the development of GFP^-^ control cells in cultures generated from PD:ibot mice *vs.* control cells in cultures generated from control mice.

Interestingly, we found that the percentages and sizes of lamellar cells in control cells in PD:ibot cultures were smaller than in control cells in control cultures (Figure 6C, D text page 21). Although both groups of cells are competent in exocytosis, they were surrounded by exocytosis-deficient *vs.* exocytosis-competent neighbor cells. The differences in the growth capacity of control cells in the presence of different neighbor cells reveal cell non-autonomous contributions of botulinum-expressing cells in oligodendrocyte development.

(3) Vamp3 levels should be examined in addition to Vamp2; otherwise, the authors cannot conclude Vamp1/2/3-dependent exocytosis is involved in the phenotypes observed.

We agree with the reviewers and examined Vamp3 levels with Western blot. We found diminished levels of Vamp3 in oligodendrocyte-lineage cells from PD:ibot mice (Figure 1J, M, text page 9).

(4) The data are convincing that botulinum-expressing oligodendrocytes do not mature properly. Further, the data support a role for L-PGDS in oligodendrocyte maturation. However, the link between these two points is currently weak, and there is a concern about correlation vs. causation in the interpretation of the data. Experiments to more conclusively show causation are required.

We agree with the reviewers that the potential link between the role of L-PGDS and the development defect of botulinum-expressing oligodendrocytes is a key question to investigate. We started preparing mouse strains for investigating this question years ago and are excited to report our new data in this revised manuscript. Our collaborators Yoshihiro Urade and Ko Fujimori’s groups in Japan generated L-PGDS overexpressing transgenic mouse strains (*Ptgds*-TG). Despite challenges in shipping live mice overseas exacerbated by the COVID-19 pandemic, we were able to cryopreserve the sperms of the TG mice, ship the sperm, and recover live mice in the U.S. We then generated triple transgenic mice (PDGFRα-creER; flox-stop-flox-botulinum toxin B-GFP; *Ptgds*-TG). Interestingly, we found that *Ptgds*-TG partially rescued the myelination defect of PD:ibot mice (Figure 9E-H, text page 29).

We further assessed whether L-PGDS can rescue the oligodendrocyte development defect of botulinum-expressing cells in purified OPC cultures in vitro to avoid the confounding effect of L-PGDS on other cell types in vivo. We found that recombinant L-PGDS protein rescued the PD:ibot oligodendrocyte development defect in vitro (Figure 9A, B, text page 27).

Thanks to an excellent suggestion by Reviewer#1, we also added HPGD, a protein that inactivates PGD2, to OPC cultures and found that HPGD inhibits oligodendrocyte maturation (Figure 7F, G, text page 26).

Together, these results strengthen the link between the role of L-PGDS and the development defect of botulinum-expressing oligodendrocytes.

Reviewer #1 (Recommendations for the authors):– For in vivo experiments examining oligodendrocyte lineage cells, the authors note that mice with only the Cre transgene or only the ibot transgene were subjected to tamoxifen. Experimental panels note "Controls." It is unclear if all these mice are congenic, which should be noted in the methods either way.

All mice used were congenic. We used PD:ibot and control littermates raised under the same maternal care in the same cage for all experiments to minimize environmental confounding factors. We added this information in the methods (text page 42).

It is also unclear if the different control genotypes were pooled for the quantifications. This should also be clarified in the methods.

We noted in the figure legends whether Cre-only or ibot-only controls were used for each experiment.

Finally, the authors note "very little GFP expression" in controls. This seems critical data to show and quantify.

We quantified the data and included a new figure (Figure 1—figure supplement 3).

– Figure 1B has a typo in the y axis label "ExpressionI".

Corrected.

– In Figure 1I, only VAMP2 expression is analyzed by Western blot. As VAMP3 is implicated in PLP trafficking and may have cell-autonomous effects in oligodendrocyte differentiation/myelination, and also has differentiation-dependent changes in gene expression from OPC to NFO/MO (Figure 1C), showing the levels of VAMP3 in PD:ibot OPCs would provide better mechanistic understanding at the effects of ibot expression.

We analyzed VAMP3 levels and found a reduction in PD:ibot cells (Figure 1J, M, text page 9).

– It is unclear what region of cerebral cortex was used for quantification throughout the manuscript. Was the same region used for all experiments? These details should be noted in the methods.

We included the information in the method (text page 47).

– In Figure 3N, the representative EM images used for quantification are too small and could stand to be enlarged to better appreciate the differences.

We enlarged the EM images.

It is unclear what criteria were used to determine myelinated axons and how g ratio was quantified. These details should be provided in the methods section.

We added the information in the Methods section (text page 49).

– Add best-fit lines to Figure 3Q.

Added.

– Figure 3R-U:– Mice used for behavioral testing are 2-5x older than all mice used for cellular quantification (P8/P17/P30 vs P60-150).– Between P8 and P30, the relative reduction in oligos is progressive. A great deal of myelin is added between 2 months and 5 months. Which data points came from which age groups? Are there age-related differences driving these results?

We have separately labeled data points from 2 months old and 5 months old mice (Figure 3Q-S). With the data we have so far (n=20-27 per genotype), there isn’t a striking progression of phenotype with age. Future analysis at multiple time points may resolve any age-dependent changes in the phenotype.

– Comparison of botox effects in OPCs vs astrocytes/endothelial cells uses different methods of expression (i.e. inducible for OPCs with CreER and constitutive with Cre for astrocytes/endothelial cells). It should at least be noted that embryonic/early postnatal compensation could partially account for the lack of effect (e.g. secreted factors/interactions from both cell types could play a role in oligodendrocyte lineage development in vivo). Additionally, repeated tamoxifen injection is known to influence oligodendrogenesis. If gfap/tie2-Cre mice were not also treated with tamoxifen, this could serve as a potential confound and should be stated as such.

We agree with the reviewer and added this point (page 15). We injected the same amount of tamoxifen to the littermates (control) of PD:ibot mice. Therefore, the developmental defect in PD:ibot mice is unlikely due to tamoxifen treatment.

– Given how critical the well-insertion experiments are in Figure 4K-M to the claims in this study, I would suggest splitting these panels and making a new figure and adding example images used for quantification from these cultures.

We made a new figure reporting the insert experiment results and added example images (Figure 6E-H).

As an alternative experiment, the authors could consider co-culturing Hpgd-overexpressing microglia along with control OPCs.

We thank the reviewer for a great idea. Inspired by the reviewer’s suggestion, we designed a simplified version of the suggested experiment. We added HPGD protein to the culture and found that HPGD inhibited oligodendrocyte maturation (Figure 7F, G, text page 26). This experiment provides additional evidence for the role of PGD2 in oligodendrocyte development.

– In Figure 5, expression of mature oligodendrocyte markers is reduced in purified oligodendrocytes using equivalent amounts of cDNA to compensate for presumably lower numbers of oligodendrocytes in the PD:ibot mice. CC1 and olig2 were used to count cell numbers by immunohistochemistry. Do these genes also show reduced expression in the RNA sequenced oligodendrocytes? Could it be that the number of oligodendrocytes counted is underestimated due to the reduced expression of selected markers used for immunostaining?

*Qk* (its encoded protein is labeled by the CC1 antibody) and *Olig2* mRNA levels did not show significant differences between PD:ibot and control purified oligodendrocytes. We added these result (Figure 4—figure supplement 1, text page 17).

– For Figure 7, gene name (Ptgds-/-) should be used in place of LPGDS-/-.

Corrected (new Figure 8).

– In the Discussion (line 622), the word "strongly" is a bit of an overstatement given the small effect size in the referenced experiment.

We removed this word.

Reviewer #2 (Recommendations for the authors):There were a number of concerns in the article in its present form.1. The notion of VAMP-dependent vesicular release from oligodendrocyte progenitor cells (OPCs) was interesting but warrants additional validation. Within neurons, there are definable sites of vesicular release and it would be particularly relevant to determine where within the OPC the tSNARE is expressed morphologically.

We agree with the reviewer that the subcellular distribution of VAMP-associate vesicles is an important question to study. We stained for VAMP2 and VAMP3 proteins in cultured OPCs and found distributed VAMP2/3 throughout the cell, including in the soma and the processes (Figure 1 —figure supplement 1, text page 5).

2. Figure 1 data show relative VAMP1, 2 and 3 expression but these data require some context to appreciate the relative abundance of these genes in relation to neurons.

We now included data on *Vamp1, 2*, and *3* expression in neurons, astrocytes, microglia, and endothelial cells in addition to OPCs and oligodendrocytes (Figure 1A-C). These comparisons show that *Vamp3* expression is higher in OPCs and oligodendrocytes than in neurons and that *Vamp2* expression in OPCs and oligodendrocytes is about 1/3 the level in neurons.

2. Figure 1 Panel D requires an indication as to where the subsequent higher magnification images are taken.

Added.

It would also be warranted to define the relative efficiencies in multiple locales, including gray matter and white matter ROIs.

We now added the quantification of efficiencies in cerebral cortex-grey mater, corpus callosum-white matter, and striatum (Figure 1G, text page 8).

3. The analysis of cell death at P30 was limited in its scope as cell death may not be limited to the oligodendrocyte lineage cells and no differences at one time point are inconclusive.

We performed a more thorough analysis of activated caspase-3 at multiple developmental stages (P8, P15, and P30) in oligodendrocytes, OPCs, and cells of other lineages. There is no significant difference between PD:ibot and control mice in any of these comparisons (Figure 3—figure supplement 1 text page 12).

4. The analysis of myelination by IHC was also considered insufficient to support the authors' conclusions that the hypomyelination phenotype is truly hypomyelination. Neurodegeneration would result in perturbed myelination during development. Analysis of axons and axon counts in ROI where hypomyelination is suggested should be performed using electron microscopy.

We performed analyses of axon densities and axon diameter in the corpus callosum region using electron microscopy. We did not detect any statistically significant differences between PD:ibot and control mice (Figure 3N, O, text page 12).

5. In many instances the data are not analyzed statistically or incompletely. For example, Figure 3Q lacks linear regression analyses and comparisons (3Q should also declare the number of myelinated axons analyzed as well).

We added linear regression and the number of axons analyzed in the figure legend (text page 14) and method (text page 49).

Figure 4M requires ANOVA for comparison across all treatment groups.

We indeed performed ANOVA for comparison across all treatment groups (*i.e.* we compared every group to every other group instead of comparing every group to one control group) with multiple comparison tests. We apologize for the lack of clarity on this point in the original manuscript. We edited the figure legend to clarify this point (New Figure 6H, text page 23).

6. The analysis of secreted factors depicted in Figure 4K-M is also poorly explained as it is described it is unclear why treatment groups 2 and 4 should differ given they both have PD:ibot and control with a permeable membrane between them.

We thank the reviewer for raising a great point. We think this result suggests that cell autonomous effect of blocking exocytosis may also contribute to the oligodendrocyte development defect in PD:ibot mice. We included the possibly contributions from both cell-autonomous and cell-non-autonomous mechanisms in Discussion (text page 35).

7. All analyses of OPC-OL differentiation based on morphology are considered suggestive but lack rigor as they are liable to subjective characterization. Additional validation using stage-specific markers is needed to support their conclusions.

We thank the reviewer for a great suggestion. We performed additional in vivo and in vitro experiments to characterize OPC differentiation with oligodendrocyte markers. We perform RNAscope experiments in vivo using probes for *Enpp6*, a marker for pre-myelinating oligodendrocytes, and *Mbp*, a marker for oligodendrocytes. Interestingly, both markers showed a significant reduction in PD:ibot mice compared with controls, suggesting that the oligodendrocyte development defect in PD:ibot mice manifest as early as the pre-myelinating stage (Figure 2I-K, text page 11). We performed quantitative real-time PCR and Western blot experiments to assess the mRNA and protein levels of MBP to assess OPC differentiation independent of morphology in vitro. We found that both MBP mRNA and protein levels are lower in PD:ibot OPC/oligodendrocyte cultures than in control cultures (Figure 5C, D, text page 19).

8. It is unclear why the authors chose to use RNAseq to evaluate secreted factors. Proteomic analysis of conditioned media would have been more appropriate and unbiased.

We agree with the reviewer that proteomics analysis of conditioned media is a more appropriate method for evaluating OPC/oligodendrocyte-secreted factors. We are performing proteomic experiments and will report the results in a separate paper in the future. We initially chose to perform RNAseq to characterize molecular changes in OPCs and oligodendrocytes in PD:ibot mice. We noticed intriguing changes of *Ptgds* and other candidate genes encoding secreted proteins. We therefore decided to assess the function of these candidate genes in oligodendrocyte development and found an interesting role of *Ptgds.*

9. The central hypothesis of the model put forth by the authors is that expression of Ptgds is a signal to promote OPC differentiation yet it is expressed across all stages of maturation in this cell lineage. Moreover, the authors contend that ptgds is expression is lower in OPCs in multiple sclerosis, while this is not entirely true since during the progressive phase of disease it is actually upregulated (PMID: 16409554). Hence, the overall premise of the correlation of expression with disease is not supported by the existing data.

We edited the text to reflect the complexity of *Ptgds* expression in multiple sclerosis based on existing literature (text page 34). In the future, comprehensive evaluation of *Ptgds* mRNA, L-PGDS protein, and PGD2 levels in different phases of multiple sclerosis (MS) will clarify potential involvement of L-PGDS in MS and the therapeutic potential of this pathway in MS.

10. Do global L-PGDS mice have a neurodegenerative phenotype?

We stained for an axon damage marker, SMI-32, and did not detect any difference between *Ptgds* knockout and littermate control mice (Figure 8H, I, text page 27). We also stained for a microglia marker, Iba1, to assess glial reactivity and did not detect any difference between *Ptgds* knockout and littermate control mice (Figure 8J, K, text page 27).

Reviewer #3 (Recommendations for the authors):(1) In order to see that the impact on differentiation and myelination in vivo is plausibly due to autocrine/paracrine-acting secreted factors, rather than a consequence of blocking of secretion itself, it is important to show whether GFP+/ibot cells differentiate and survive in vivo. The authors should show quantification of the fraction of oligodendrocytes with MBP+ sheaths that are GFP+ or GFP- (this can be done in the cortex where myelin is sparse) or CC1+ cells if needed.

We analyzed the overlap between GFP^+^ botulinum-expressing cells and the population of differentiated oligodendrocytes (Olig2^+^PDGFRa^-^CC1^+^) and found that botulinum-expressing cells can survive and become differentiated oligodendrocytes (Figure 3—figure supplement 2, text page 12).

(2 and 3) The authors should confirm with an MBP staining-independent membrane label (not cytoplasm) the conclusions on lamellar cell morphology. Otherwise, the fraction of MBP+ cells may be a more accurate reflection of the data.

Completed as suggested, described above.

Additional concern:I'm not sure whether this is an artifact introduced in the creation of the pdf (why I've included this as a private recommendation), but there are some issues with the images in Figures 2 and 7. Several images have clear boundaries with differing background intensities within the single image. This makes the images appear as they have been pasted together from separate images and/or brightness/contrast was not uniformly treated across the entire image. How were the original images modified? Are these stitched images with different acquisition settings?Specifically, in 2F: The first two Olig2 images have a clear boundary in the images. In the PDGFRa image on the right-hand side, there is an apparent "box" in the lower-left (boundary and different background intensities).Specifically in 7: A and C have variations in the background intensities in tile-like patterns within the images.If the high-resolution versions of these images lack this issue, please ignore! If not, I ask the authors please address this.

We thank the reviewer for pointing this out. We typically image many fields of view using our Zeiss Apotome widefield fluorescent microscope and stitch the tiled images together into a larger image using an automatic stitching function in the Zeiss Zen software. For some reason, the background fluorescence intensity is not even across each tile, and we often see clear boundaries with differing background intensities where different tiles are stitched together. We confirm that we only stitched images from adjacent fields of view together and we always uniformly adjusted brightness and contrast across the entire image. We have included images without clear boundaries in this revised manuscript.